# DeepASA: An Object-Oriented Multi-Purpose Network for Auditory Scene Analysis

**Dongheon Lee, Younghoo Kwon, Jung-Woo Choi**
Korea Advanced Institute of Science and Technology (KAIST)
Daejeon, Republic of Korea
{donghen0115, k0hoo, jwoo}@kaist.ac.kr

## Abstract

We propose DeepASA, a multi-purpose model for auditory scene analysis that performs multi-input multi-output (MIMO) source separation, dereverberation, sound event detection (SED), audio classification, and direction-of-arrival estimation (DoAE) within a unified framework. DeepASA is designed for complex auditory scenes where multiple, often similar, sound sources overlap in time and move dynamically in space. To achieve robust and consistent inference across tasks, we introduce an object-oriented processing (OOP) strategy. This approach encapsulates diverse auditory features into object-centric representations and refines them through a chain-of-inference (CoI) mechanism. The pipeline comprises a dynamic temporal kernel-based feature extractor, a transformer-based aggregator, and an object separator that yields per-object features. These features feed into multiple task-specific decoders. Our object-centric representations naturally resolve the parameter association ambiguity inherent in traditional track-wise processing. However, early-stage object separation can lead to failure in downstream ASA tasks. To address this, we implement temporal coherence matching (TCM) within the chain-of-inference, enabling multi-task fusion and iterative refinement of object features using estimated auditory parameters. We evaluate DeepASA on representative spatial audio benchmark datasets, including ASA2, MC-FUSS, and STARSS23. Experimental results show that our model achieves state-of-the-art performance across all evaluated tasks, demonstrating its effectiveness in both source separation and auditory parameter estimation under diverse spatial auditory scenes. The demo video, samples and code are available at https://huggingface.co/spaces/donghoney22/DeepASA.

## 1 Introduction

Auditory scene analysis (ASA) seeks to extract information about individual sound sources from complex auditory environments [1]. This includes identifying the class of each source, its onset and offset times, and its direction of arrival (DoA) [2]. In humans, ASA is facilitated by the brain's ability to organize sound into perceptual streams by integrating multiple auditory cues such as pitch, timing, and spatial location, in a complementary fashion [3, 4, 5].

Inspired by this ability, computational auditory scene analysis (CASA) research [2, 6] has aimed to replicate similar functionality using auditory cues like pitch, onset/offset, interaural level differences (ILDs), and interaural time differences (ITDs). With the rise of deep learning, a wide array of models has emerged to tackle specific ASA tasks, including audio tagging (AT) [7, 8, 9], sound event detection (SED) [10, 11, 12], DoAE [13, 14, 15], and blind source separation (BSS) [16, 17, 18].

However, unlike the human auditory system, these task-specific models typically lack the capacity for relational reasoning across tasks and cues. As a result, they often fail when critical auditory cues

are missing or degraded [19]. Recent work highlights the benefit of combining multiple auditory cues. For instance, target sound extraction (TSE), which uses class [20, 21], activation [22, 23], and spatial cues [24, 25], outperforms universal sound separation (USS) in challenging scenarios [26, 27]. Similarly, joint SED and DoAE in sound event localization and detection (SELD) has been shown to enhance performance by leveraging interdependent auditory information [28, 29].

Building on this evidence, we propose a general-purpose ASA model that emulates this cue integration process. Our model separates object-level auditory streams at an early stage and performs multiple downstream ASA tasks by exploiting the complementary relationships among estimated cues.

**Contribution.** We introduce DeepASA, a unified architecture for general auditory scene analysis, with two core contributions:

1. **Object-Oriented Processing (OOP)**: We design an encoder-decoder framework that isolates features of individual sound objects. The encoder incorporates a dynamic short-time Fourier transform (STFT) module and a transformer-based feature aggregator. These are followed by an object feature separator that disentangles per-object auditory representations. These object-centric features are then passed to multiple sub-decoders, each responsible for estimating a specific auditory parameter (e.g., class, DoA, activation) without requiring manual association or permutation invariant training between estimated auditory parameters.

2. **Chain-of-Inference (CoI) architecture**: To emulate the human auditory system's ability to recover missing cues by reasoning from others, we propose a chain-of-inference architecture. This component progressively refines object representations by fusing the outputs of ASA sub-decoders using a temporal coherence-based attention mechanism. It enables the model to reinforce incomplete or ambiguous estimates via complementary information.

We validate DeepASA on multiple spatial audio benchmarks. Ablation studies on the newly proposed ASA2 dataset[1] show that task-specific sub-decoders mutually enhance one another when conditioned on shared object features, yielding strong results with an SI-SDRi of 11.2 dB and a SELD score of 0.206. We also demonstrate that DeepASA pretrained by the ASA2 dataset can be generalized to various spatial audio benchmarks through fine-tuning with CoI. The fine-tuned DeepASA achieves state-of-the-art (SOTA) results, including an SI-SDRi of 18.5 dB on MC-FUSS for multichannel universal sound separation [30, 31], a SELD score of 0.253 on STARSS23 [32].

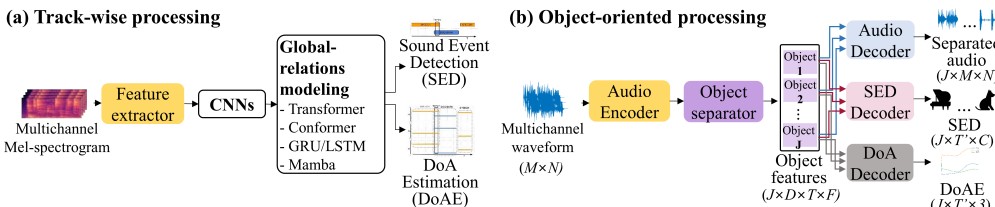

Figure 1: Comparison between (a) traditional track-wise processing and (b) proposed object-oriented processing

## 2 Method

### 2.1 Overall framework: DeepASA

The multichannel audio mixture $\mathbf{x} \in \mathbb{R}^{M \times N}$, where $M$ is the number of microphones and $N$ is the number of time samples of the waveform, can be modeled as the summation of $J$ reverberant foreground signals and background noise $\mathbf{v}$. The $j$-th foreground source can be further decomposed as the superposition of direct sound $\mathbf{s}_j$ and reverberation $\mathbf{h}_j$.

$$\mathbf{x}[n] = \sum_{j=1}^{J} (\mathbf{s}_j[n] + \mathbf{h}_j[n]) + \mathbf{v}[n]. \tag{1}$$

[1] https://huggingface.co/datasets/donghoney22/ASA2_dataset

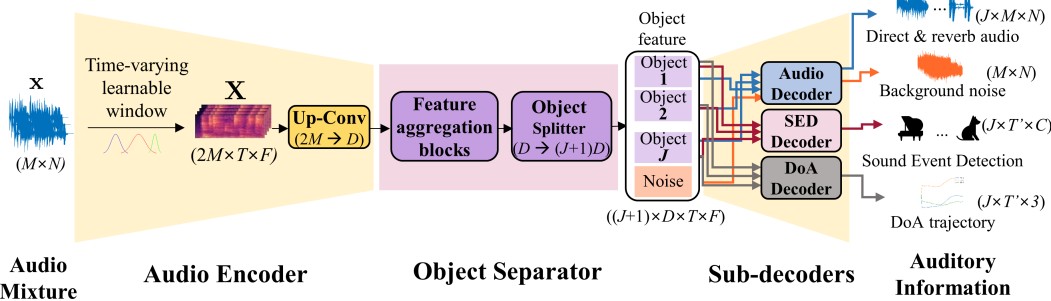

Figure 2: Basic architecture of DeepASA

We aim to estimate the auditory information of up to $J$ foreground sources, including classes, activations of onsets and offsets, DoA trajectories, and multichannel waveforms of the direct and reverb audio signals, as well as one multichannel noise signal, from the multichannel audio mixture.

As illustrated in Figure 2, the overall architecture of DeepASA comprises three components: audio encoder, object separator, and sub-decoders. The audio encoder extracts basic features from the $M$-channel input mixture. The object separator separates audio features extracted from the audio encoder into $J + 1$ object features, each of which encapsulates the auditory information of a single sound object. The sub-decoders then estimate the auditory information from each object feature. $D, T'$ and $C$ are denoted as channel dimension, time frames and number of classes.

The proposed framework separates each sound object at a feature level, which is referred to as object-oriented processing (OOP) in this work. One advantage of OOP is that the object-wise permutation of the separated features is consistently inherited across all sub-decoders. For example, the $j$-th object estimated by the audio, SED, DoA decoders all include the waveforms, SED, DoA information from the same sound source. The characteristic is the major difference to the conventional track-wise processing [28, 33, 34, 35, 36, 37], for which multiple source information can be aligned on the same track when they do not overlap in time. The OOP eliminates the requirement for pairing different auditory information and also enables the selective attention of estimated information across different objects and auditory information.

## 2.2 Audio encoder

**Dynamic STFT with time-varying learnable window** The audio encoder of DeepASA incorporates our proposed architecture, Dynamic STFT, for adaptive temporal focusing on the input waveform. Conventional STFT uses a fixed analysis window at all times, yet the locations and durations of salient information within time frames can vary across the waveform. Accordingly, we employ a learnable Gaussian window whose mean ($\mu_t$) and standard deviation ($\sigma_t$) are predicted at every frame, forming $\boldsymbol{\mu} \in \mathbb{R}^T$ and $\boldsymbol{\sigma} \in \mathbb{R}^T$ vectors for the total number of time frames $T$. The frame-wise mean ($\mu_t$) aligns the center of the analysis window on the most informative region, while ($\sigma_t$) sculpts its taper: a large $\sigma_t$ flattens the analysis window toward a rectangle, sharpening spectral focus with a thin main lobe, whereas a small $\sigma_t$ contracts it toward an impulse, enhancing temporal focus.

The process of extracting $\boldsymbol{\mu}$ and $\boldsymbol{\sigma}$ is depicted in Figure 3. For each microphone channel, a parameter-shared 1D convolution with the same kernel size and stride as the window size $K$ and hop size $L$ used in STFT is applied, producing outputs $\mathbf{x}' \in \mathbb{R}^{D \times T}$ for each microphone. Two parallel linear layers then project $\mathbf{x}'$ to the mean and standard deviation, producing $\boldsymbol{\mu}, \boldsymbol{\sigma} \in \mathbb{R}^T$ for every microphone channel.

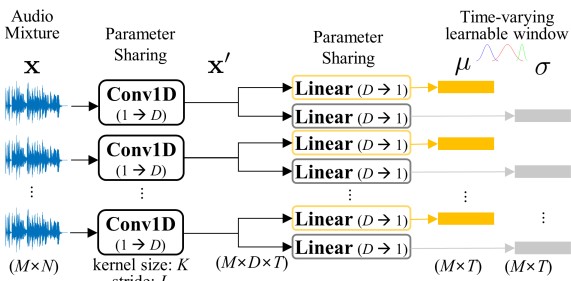

Figure 3: Time-varying learnable window

Processing STFT with the time-varying learnable window, the multichannel waveform is converted into a complex spectrogram $\mathbf{X} \in \mathbb{R}^{2M \times T \times F}$ where $F$ is the number of frequency bins. These

complex spectrograms are then fed directly into an Up-Conv block composed of a 2D convolution layer followed by layer normalization, to increase the channel dimension from $2M$ to $D$ while preserving the temporal and spectral dimensions. For training a learnable window, the parameters are frozen at the beginning of the training, and when the model has converged, they are unfrozen and trained together.

## 2.3 Object separator

**Feature aggregation** As the feature aggregation block of the proposed network, we modified the DeFT-Mamba [38] model designed for the USS task (Figure 4(a)). DeFT-Mamba is a comprehensive feature aggregation model utilizing Mamba and transformer layers to capture temporal, spectral, and interchannel relations in multidimensional data. To lighten the model, we adopt Mamba-FFN only for T-Hybrid Mamba and use a conventional feedforward network (FFN) for F-Hybrid Mamba. Then, we remove the unfolding process from the gated convolutional block (GCB) while keeping the gating module and the convolution kernel.

**Object splitter** The features arranged by the modified DeFT-Mamba are transformed into $J + 1$ object features through the object splitting layer given by a 2D convolution kernel. The separated $J$ object features corresponding to foreground sources pass through the direct and reverb decoders of the audio decoder, the SED decoder, and the DoA decoder. The last object among the separated objects is treated as a noise object, which is separately utilized by the noise decoder in the audio decoder to estimate background noise.

## 2.4 Sub-decoders

The sub-decoders decode each object feature into the direct, reverberant, and noise audio signals, class, activation, and DoA outputs. The detailed architectures of sub-decoders are depicted in Figure 4(b).

**MIMO Audio decoder** The audio decoder aims to estimate a maximum of $J$ foreground source signals and 1 background noise signal corresponding to individual sound objects. In the conventional multi-input single-output (MISO) model, spatial information is lost in the decoded single-channel signal, and the audio decoder cannot assist DoAE in the DoA decoder. Therefore, the audio decoder in DeepASA is designed as a multi-input multi-output (MIMO) architecture, facilitating spatial information in the separated object features. To further assist the development of spatial information and dereverberation, the audio decoder is trained to separate direct sound $\mathbf{s}_j$ and reverberant sound $\mathbf{h}_j$ of each object across all microphone channels. In addition, for reducing the influence of background noise, the last object from the object separator is separately processed by the noise decoder trained to estimate the background noise $\mathbf{v}$. When active foreground sources are fewer than $J$, inactive sources are trained to estimate zero target signals.

**SED decoder** The SED decoder decodes the object features of each sound source into the predicted class probability ($1 \times C$), the binary object activation curve for event onset and offset ($T' \times 1$), and the predicted SED map ($T' \times C$). Here, $T'$ and $C$ denote the number of time frames and the number of classes, respectively. The predicted SED map is only used in the training stage to guide

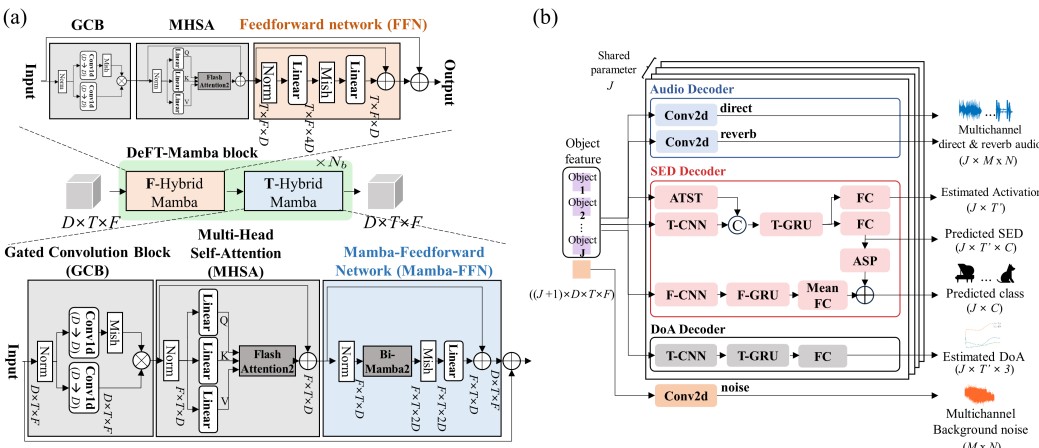

Figure 4: Detailed architecture of (a) feature aggregation block, and (b) sub-decoders

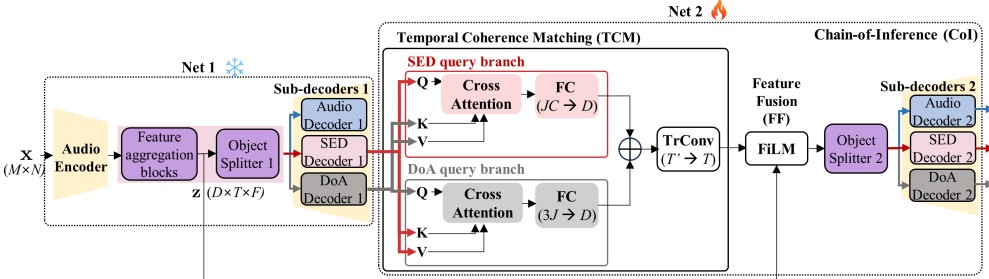

Figure 5: Detailed architecture of chain-of-inference

the joint detection of a sound event and class. In the inference stage, we use the SED map produced by multiplying the predicted class and activation curve, resulting in a single SED map for each sound source. This is to exclude the possibility of more than one class being mapped to the SED map for each sound object. The SED decoder combines a pre-trained audio teacher-student transformer (ATST) [12] with two separate convolutional recurrent neural network (CRNN) [11]-based branches summarizing time and frequency information. CRNN consists of multiple repetitions of Conv2d and Maxpool2d (denoted as CNN), followed by GRU and FC layers. Specifically, among the three red branches of the SED decoder, the middle path with seven Conv+Maxpool layers outputting $(J \times T' \times D)$ is the T-CRNN path, and the bottom path with two Conv+Maxpool layers outputting $(J \times F' \times D)$ corresponds to the F-CRNN path. We employ the pre-trained ATST to utilize various features learned from AudioSet, a large-scale dataset containing a wide range of audio classes [39]. The features from ATST are combined with auxiliary features developed from another Conv2d layer. The concatenated features are analyzed in time by Time (T)-GRU and summarized to predict the sound event map. For predicting a single-channel activation curve, adaptive statistical pooling (ASP) [40] and one fully connected layer are applied along the class dimension. The class probability is predicted by superposing the logit derived from Frequency (F)-CRNN, which prioritizes the spectral information using Conv2d layers and F-GRU, with the logit obtained from ASP. For inactive sources, the class decoder is trained to predict an equal probability of $1/C$ across all $C$ classes, while the sound event and activation decoders are trained to produce zero values for all time frames. The detailed specifications of the SED decoder are provided in Appendix A.

**DoA decoder** The DoA decoder has the CRNN structure and outputs a stream of DoA vectors in Cartesian coordinates $(x, y, z)$. The DoA output estimates a DoA vector with a magnitude of 1 when a source is present, and assigns all $x$, $y$, $z$ values to 0 when no source is detected, allowing for the estimation of the activity and incident angle. DoA is estimated for each time frame, enabling the trajectory prediction for moving sources.

## 2.5 Chain-of-inference

The initial estimation through sub-decoders can include misalignment between auditory parameters. For example, the DoA vector stream can exhibit different onsets and offsets with the sound event map. These asynchronous estimations are refined from the object separation level through the fusion of initial estimations. This refinement step, chain-of-inference, consists of two processing steps: temporal coherence matching (TCM) and feature fusion (FF) (Figure 5).

**Temporal coherence matching** In TCM, temporal coherence between SED and DoAE is evaluated through the multi-clue attention mechanism. One branch of multi-clue attention takes SED as queries and DoA as keys and values, and the attention output is added to SED to refine the SED information. Conversely, the other multi-clue attention branch uses DoA as queries and SED as keys and values. The SED decoder outputs three components, but ultimately, the SED map of $T' \times C$ is constructed for each object from the predicted class probability $(1 \times C)$ and object activation $(T' \times 1)$. DoA decoder also generates T'x3 output for each object. The alignment between the SEDs and DoAs is accomplished in TCM through the cross-attention by assigning the time dimension as a sequence dimension. The outputs of the two multi-clue attentions are passed through two individual linear layers to produce the $D$-channel clues and then superposed for fusion. The fused clue is interpolated by a transposed convolution layer to match its time dimension $T'$ with that of object features $(T)$.

**Feature fusion** In the feature fusion block, the fused clue is utilized to refine the object feature separation. TCM output is processed by a feature-wise linear modulation (FiLM) layer [41], which outputs values of $\beta$ and $\gamma$ for each channel and time frame. The output of the FiLM layer is then combined with the output from the feature aggregation module ($\mathbf{z}$) to inject information required for object separation. The object features refined by the object splitter 2 are then plugged again into the sub-decoders 2 to improve ASA parameter estimation.

For efficient training of the CoI architecture, we utilize multi-stage training. That is, Net 1 of Figure 5 is trained first, then Net 2 is trained with up to the third DeFT-Mamba block of Net 1 frozen.

## 3 Experimental Settings

### 3.1 Datasets

The proposed architecture is trained and evaluated on three spatial audio datasets: ASA2, MC-FUSS, and STARSS23. Auditory Scene Analysis dataset V2 (ASA2) provides ground truth signals and labels for all acoustic parameters addressed in this study, including comprehensive auditory information for moving source separation and scene analysis in noisy, reverberant environment. ASA2 is based on the ASA dataset [38], but we modified it extensively to enable the comprehensive tasks discussed in this study. ASA2 supports (1) the separation of direct and reverberation components (for dereverberation), (2) MIMO separation, as compared to MISO separation of the ASA dataset, and (3) an increased maximum number of foreground sound sources, from four to five. The balance in the number of audio clips has been changed to be proportional to the number of sources constituting the scene. Multichannel-Free Universal Sound Separation (MC-FUSS) [30] is a benchmark dataset for multichannel USS, including mixtures of 1–4 sound sources, with one being the background source. For a fair comparison on the MC-FUSS dataset, we trained DeepASA both from scratch and with fine-tuning of the pre-trained model on the ASA 2 dataset. Also, we compared the effect of pre-trained ATST in the MC-FUSS dataset. STARSS23 [32] is a dataset for the SELD task, addressing various sound events with 3.8 hours of audio-video data collected from real environments. This dataset generally requires data augmentation, so additional data augmentation of 80 hours was conducted using a spatial scaper [42]. In this experiment, other SELD models were also trained using the large auxiliary datasets. The DCASE 2023 challenge allowed the use of external datasets and pre-trained models, so we applied this rule. All datasets were set to a 16 kHz sampling rate and a 4-second duration.

### 3.2 Training and model configuration

For the STFT, the Gaussian window of 40 ms length and hop size of 20 ms (50% overlap) was used. The initial weight for the mean and standard deviation of the window was set to 0 and 6.67 ms, ensuring $6 \times \sigma$ covers the full window range. The number of feature aggregation modules was 6, and the channel dimension for the DeepASA was 64. The initial learning rate $4 \times 10^{-4}$ was used for the Adam optimizer during pre-training, and $4 \times 10^{-4}$ and $5 \times 10^{-5}$ were used for fine-tuning to the MC-FUSS and STARSS23 datasets, respectively. The learning rate was halved if the validation loss did not decrease for five consecutive epochs. The model was trained for 100 epochs, and the model from the epoch with the lowest validation loss was selected as the final model. The batch size was set to 2. The training was performed on eight GeForce RTX 3090 GPUs.

The direct and reverb decoders in the audio decoder were trained with the source-aggregated signal-to-distortion ratio (SA-SDR) loss [43] with permutation invariant training (PIT) [44]. We configured a reference channel among multichannel signals and applied different weights (1:0.1) to the losses of the reference and the other channels. The noise decoder was trained by the scale-invariant signal-to-distortion ratio (SI-SDR) loss [45]. We employed one noise decoder, so the noise decoder was not trained with PIT. The joint loss was calculated by summing the losses with weights 1:1:0.01 for direct, reverb, and noise decoders. The losses for the SED decoder were cross-entropy, binary cross-entropy, and mean-square-error (MSE), for the class decoder, activation decoder, and sound event decoder, respectively. The MSE loss was used for the DoA decoder, which was summed with all aforementioned losses with equal weights. In evaluation, scale-invariant signal-to-distortion ratio improvement (SI-SDRi) (dB) and signal-to-distortion ratio improvement (SDRi) (dB) were used for USS performance evaluation, while SELD metrics (error rate (ER,%), F1-score (F1,%),

localization error (LE, degrees), localization recall (LR, %)) [28] were employed for SED and DoAE. We measured the model complexity by the parameter size and computational complexity.

# 4 Results

## 4.1 Ablation study of DeepASA blocks

We conducted ablation studies to assess the contribution of each module in DeepASA. Each row block of Table 1 corresponds to the ablation of a specific component. For sequential consistency, the final configuration from the previous block was used as the baseline for the subsequent analysis.

**Framework** The first ablation study is about incorporating the MIMO separation task, SED, and DoA decoders. The baseline was the MISO USS model [38], and USS performance was measured based on the reconstruction of the reference channel. When MIMO USS is introduced, the separation performance of the reference channel decreases. This can be attributed to the features extracted for all channels rather than a single channel. Introducing the SELD task using the SELDNet [28] sub-decoder enhances the USS performance as well, showing that auditory scene analysis can effectively guide the source separation task. The SELD performance is further improved when the CRNN architectures are employed as SED and DoA decoders. Notably, USS performance metrics are recovered to those of the baseline, indicating that a sophisticated SELD decoder can effectively improve SELD performance with minimal impact on the USS task.

**Object separator** Next, we examine the impact of downsizing on the baseline model (DeFT-Mamba-MIMO + SED, DoA decoder). The performance reduction by omitting unfold was negligible, showing only -0.1 dB difference in SI-SDRi. Furthermore, when F-Mamba is removed, a performance degradation of 0.3 dB in SI-SDR and 0.002 in SELD score is observed. However, this comes with significant lightweighting in parameter size (0.9 M) and computational complexity (10.9 GMac/s), so we adopted the conventional FFN in the F-HybridMamba block.

**SED decoder** The efficacy of more sophisticated sub-decoders is tested in this section. Recent work has shown that combining pre-trained ATST and CRNN achieves high performance in sound event detection [46]. The SELD score is also significantly improved when the baseline, combined with the ATST + T-CRNN decoder, is fine-tuned to the ASA2 dataset. This stresses the importance of using the class decoder specialized to the classification task. Similarly, we combined the F-CRNN architecture to capture and emphasize global frequency relations, improving the SELD score further up to 0.266. Since the improvement was less only with the F-CRNN decoder, we can conclude that capturing both local and global time-frequency relationships is critical for effective SED.

**Audio decoder** With the best SED decoder suggested above, we conducted the model analysis for the audio decoder. The first improvement was adding a noise decoder that estimates multichannel background noise. Both separation and SED performance are significantly improved when the noise decoder performing noise estimation is added. As analyzed in detail in Appendix B, the noise decoder enhances the separation performance of domestic sounds that share similar time-frequency

Table 1: Ablation study on the proposed model using the ASA2 dataset. The models with (+) or (-) indicate the addition or removal of the corresponding blocks from the baseline. The (+) in the parameter size indicates the parameter size of the pre-trained ATST.

| | Model variation | USS | | SED | | DoAE | | SELD ↓ | Complexities | |
|---|---|---|---|---|---|---|---|---|---|---|
| | | SI-SDRi ↑ | SDRi ↑ | ER ↓ | F1 ↑ | LE ↓ | LR ↑ | | Param. | MAC/s |
| Framework | DeFT-Mamba-MISO [38] | 10.4 | 11.3 | - | - | - | - | - | **3.6 M** | **83.8 G** |
| | DeFT-Mamba-MIMO | 10.0 | 10.9 | - | - | - | - | - | 3.6 M | 83.8 G |
| | (+) SELDNet [28] | 10.2 | 11.1 | 42.0 | 58.2 | 28.6 | 63.2 | 0.341 | 5.4 M | 86.1 G |
| | (+) SED, DoA decoder | 10.4 | 11.4 | 40.0 | 60.3 | 22.9 | 65.7 | 0.317 | 7.2 M | 88.4 G |
| Object separator | (-) Unfold | 10.3 | 11.4 | 39.8 | 60.5 | 22.2 | 66.1 | 0.314 | 7.2 M | 88.0 G |
| | (-) Unfold, F-Mamba | 10.0 | 11.1 | 39.9 | 60.4 | 22.0 | 65.9 | 0.315 | 6.3 M | 77.1 G |
| SED decoder | (+) F-CRNN | 10.3 | 11.2 | 38.0 | 61.1 | 21.9 | 68.8 | 0.301 | 8.1 M | 90.7 G |
| | (+) ATST + T-CRNN [46] | 10.2 | 11.2 | 35.7 | 66.1 | 21.2 | 71.0 | 0.276 | 6.3 (+96.8) M | 96.5 G |
| | (+) ATST + T- &F-CRNNs | 10.3 | 11.2 | 34.1 | 66.6 | 21.3 | 72.8 | 0.266 | 8.1 (+96.8) M | 98.8 G |
| Audio decoder | (+) Noise decoder | 11.0 | 11.7 | 30.3 | 69.8 | 21.2 | 76.0 | 0.241 | 8.1 (+96.8) M | 98.9 G |
| | (+) Direct/reverb, noise decoder | 10.8 | 11.5 | 30.0 | 69.6 | 19.1 | 76.2 | 0.237 | 8.1 (+96.8) M | 99.0 G |
| Dynamic STFT | (+) Time-invariant window | 10.7 | 11.4 | 30.2 | 69.8 | 19.3 | 76.0 | 0.238 | 8.1 (+96.8) M | 99.0 G |
| | (+) Time-variant window | 11.0 | 11.7 | 28.8 | 70.2 | 18.5 | 76.9 | 0.230 | 8.2 (+96.8) M | 99.1 G |
| **CoI** | **(+) Chain-of-inference** | **11.2** | **12.0** | **25.0** | **74.1** | **17.0** | **78.1** | **0.206** | 12.1 (+96.8) M | 104.0 G |

characteristics with background noise. Next, separate decoders for the direct and reverberant sounds were employed to estimate $\mathbf{s}_j$ and $\mathbf{h}_j$, respectively, instead of $\mathbf{s}_j + \mathbf{h}_j$ estimated by the baseline audio decoder. Here, the USS performance was evaluated by measuring SI-SDRi and SDRi on $\mathbf{s}_j + \mathbf{h}_j$ for fair comparison to previous cases. While this change slightly reduces separation performance, DoAE performance (LE and LR) is markedly improved. This suggests that learning to separate direct audio positively impacts DoA accuracy by suppressing reverberant signals. A detailed analysis of the direct/reverb decoders is provided in Appendix C.

**Dynamic STFT** Lastly, the effectiveness of dynamic STFT was validated. The first comparison set is the model using a learnable but time-invariant window, which was designed by applying mean pooling over the temporal dimension before passing it through the linear layers of a time-varying learnable window. This model shows negligible performance improvement compared to the baseline STFT-based model. In contrast, the time-varying learnable window increases all performance metrics, demonstrating the importance of dynamic temporal adjustment of the window position and length in feature extraction and separation. The detailed comparisons with various dynamic convolution kernels are presented in Appendix D.

## 4.2 Ablation study of chain-of-inference (CoI)

The results of the CoI ablation study are presented in Table 2. The baseline model corresponds to the best-performing configuration without CoI, identified in Table 1. To isolate the effect of individual auditory cues, we ablated the CoI module by including only a single query branch, either the SED or DoA branch in the Temporal Coherence Matching (TCM) module. Additionally, the audio decoder 2 was removed from Net 2 to eliminate the influence of two-stage training on the audio separation task. In these ablated configurations, USS performance was evaluated using the output from audio decoder 1 only.

Table 2: Ablation study for the chain-of-inference architecture

| Model variation | | USS | | SED | | DoAE | | SELD ↓ | Complexities | |
|---|---|---|---|---|---|---|---|---|---|---|
| | | SI-SDRi ↑ | SDRi ↑ | ER ↓ | F1 ↑ | LE ↓ | LR ↑ | | Param. | MAC/s |
| without chain-of-inference | | 11.0 | 11.7 | 28.8 | 70.2 | 18.5 | 76.9 | 0.230 | 8.2 (+96.8) M | 99.1 G |
| without audio decoder 2 | (+) SED branch | 11.0 | 11.8 | 26.6 | 71.8 | 18.2 | 76.5 | 0.221 | 10.3 (+96.8) M | 101.6 G |
| | (+) DoA branch | 11.0 | 11.8 | 28.2 | 70.6 | 17.6 | 76.0 | 0.228 | 10.3 (+96.8) M | 101.6 G |
| | (+) SED & DoA | 11.0 | 11.7 | 26.5 | 73.0 | 17.3 | 77.2 | 0.214 | 12.1 (+96.8) M | 104.0 G |
| + Chain-of-inference | | $11.2_{\pm0.1}$ | $12.0_{\pm0.1}$ | $25.0_{\pm0.4}$ | $74.1_{\pm0.3}$ | $17.0_{\pm0.3}$ | $78.1_{\pm0.4}$ | $0.206_{\pm0.001}$ | 12.1 (+96.8) M | 104.0 G |

The results reveal that incorporating the SED branch improves sound event detection (as indicated by enhanced ER and F1-score), while incorporating the DoA branch improves DoA estimation (as shown by reduced localization error, LE). These findings suggest that cross-attention with SED or DoA queries selectively refines the corresponding auditory information. Notably, combining both branches (SED & DoA) leads to further improvements in both metrics, indicating that when one cue is unreliable, the other can compensate to support more accurate inference. In the final configuration, the full CoI architecture was restored, with Net 2 including audio decoder 2. In this setting, the USS output from audio decoder 2 was used for evaluation. The complete CoI architecture yields the best overall performance, achieving 12.0 dB SI-SDRi on the USS task and a SELD score of 0.206. These results confirm that the proposed CoI mechanism effectively enhances estimation across auditory domains by explicitly modeling interdependencies and enabling complementary cue integration. We run five trials and report the average and standard deviation for the final model. A more detailed analysis of CoI's contribution is provided in Appendix E

## 4.3 Generalization to spatial audio benchmarks & comparison with SOTA models

We evaluated the generalization capability of DeepASA on two spatial audio benchmarks: MC-FUSS for USS and STARSS23 for SELD, and compared it to SOTA models. DeepASA was pre-trained on the ASA2 dataset and fine-tuned using the ground-truth parameters available in each target dataset. Since no existing model can simultaneously perform USS, SED, and DoAE tasks, comparison models were trained individually for the subset of tasks they support.

**USS performance comparison on MC-FUSS dataset** Table 3 presents a comparison between DeepASA (excluding the CoI, SED, and DoA decoders) and existing SOTA models for universal sound separation (USS). Despite being unable to utilize the CoI module during fine-tuning on MC-FUSS, DeepASA outperforms existing USS models, particularly in scenes with a larger number

of foreground sources. This performance gain is largely attributed to the noise decoder's explicit estimation of background noise and the benefits of fine-tuning from pre-trained weights.

Table 3: USS performance comparison on MC-FUSS dataset

| Training | Model | J=2 | J=3 | J=4 | Total | Param. | MAC/s |
|---|---|---|---|---|---|---|---|
| From scratch | ByteDance-uss [26] | 14.8 | 14.4 | 12.7 | 14.0 | 28.0 (+80.7) M | 40.1 G |
| | MC-BSRNN [25] | 15.7 | 15.2 | 11.4 | 14.1 | 12.2 M | 15.3 G |
| | TF-GridNet [47] | 17.2 | 16.1 | 12.5 | 15.3 | 14.7 M | 462 G |
| | DeFTAN-II [48] | 17.6 | 16.3 | 12.8 | 15.6 | 4.1 M | 66.1 G |
| | SpatialNet [49] | 17.8 | 16.5 | 13.1 | 15.8 | 7.3 M | 71.8 G |
| | DeFT-Mamba [38] | 18.4 | 17.1 | 13.8 | 16.4 | 3.6 M | 83.8 G |
| | **DeepASA (SEP, w/o ATST)** | 18.5 | 18.2 | 15.7 | 17.5 | 2.9 M | 77.3 G |
| Fine-tuning (pre-trained on ASA2) | DeFT-Mamba [38] | 18.4 | 17.1 | 14.1 | 16.6 | 3.6 M | 83.8 G |
| | **DeepASA (SEP, w/o ATST)** | 18.8 | 19.0 | 17.4 | 18.4 | 2.9 M | 77.3 G |
| | **DeepASA (SEP)** | 18.9 | 19.1 | 17.6 | 18.5 | 2.9 M | 77.3 G |

**SELD performance comparison on STARSS23 dataset** We evaluate two versions of DeepASA (with and without CoI) against SOTA SELD models, including top-ranking submissions from the DCASE 2023 challenge. DeepASA achieves the highest overall performance on the STARSS23 dataset, even without employing the audio decoder. Both versions of DeepASA surpass the performance of the top-ranked model [37], which relied on class-dependent separation networks and model ensembling. It is worth noting that DeepASA was pre-trained on the ASA2 dataset prior to fine-tuning. To control for the effect of pre-training, we further compared DeepASA to publicly available models that were also trained using ASA2. As detailed in Appendix F, DeepASA consistently achieves SOTA performance across all evaluated tasks under the same training conditions. These results collectively demonstrate the strength of DeepASA across diverse ASA tasks and datasets, validating the effectiveness of combining class, activation, and DoA trajectory information through the object-oriented processing (OOP) and chain-of-inference (CoI) architecture.

Table 4: SELD performance comparison on STARSS23 dataset

| Model | ER↓ | F1↑ | LE↓ | LR↑ | SELD↓ |
|---|---|---|---|---|---|
| CST-former [34] | 59.0 | 42.6 | 20.5 | 61.3 | 0.416 |
| MFF-EINV2 [50] | 54.0 | 42.5 | 18.7 | 62.6 | 0.398 |
| CST-former2 [51] | 42.0 | 59.7 | 15.6 | 68.4 | 0.301 |
| EINV-2 w/ data augmentation chain (2nd rank) [35] | 42.0 | 57.5 | 15.8 | 72.7 | 0.301 |
| NERC-SLIP System (1st rank) [37] (w/o Ensemble) | 40.0 | 64.0 | 13.4 | 74.0 | 0.277 |
| NERC-SLIP System (1st rank) [37] (w/ Ensemble) | 38.0 | **66.0** | 12.8 | **75.0** | 0.260 |
| **DeepASA (SELD)** | 34.4 | 62.6 | 10.2 | 73.8 | 0.259 |
| **DeepASA (SELD) + chain-of-inference** | **33.7** | 63.1 | **9.8** | 74.6 | **0.253** |

# 5  Related Works

**Universal Sound Separation (USS)** USS addresses the challenge of separating various sound sources in an auditory scene where multiple sources overlap [18, 31]. Humans perform source separation by leveraging various characteristics of individual sources, such as class, activation, and DoA. However, existing USS methods rely on time-frequency representations, which may limit their ability to exploit such source-specific attributes [26, 25]. To address this, the proposed method draws inspiration from human segregation processing and integrates multiple audio cues within a unified framework, resulting in improved performance on the USS task.

**Target Sound Extraction (TSE)** TSE refers to the separation of specific sound sources from audio mixture, with a focus on the auditory information of the source [20, 21]. TSE plays a significant role in the source separation area, where spatial and temporal characteristics are leveraged to extract the target sound [22, 26]. A limitation of TSE is that humans must explicitly provide clues such as class, activation, and DoA. However, the network itself needs to iteratively infer these clues and use them to refine the separation process. To overcome this limitation, the proposed method introduces a strategy in which the clues inferred by the network are integrated to improve separation performance.

**Sound Event Localization and Detection (SELD)** The SELD task involves performing both SED and DoAE within a single model [28, 33, 51]. Nevertheless, most previous SELD models first analyze auditory cues using separate DNN branches for SED and DoA, and then perform track-wise separation into auditory stream [52, 53, 54, 50, 55]. This approach can lead to incorrect pairing of SED and DoA outputs, and especially causes performance degradation when handling polyphonic

audio. Additionally, models like [37] have improved performance by incorporating a pre-trained target sound extraction model [17] but cannot reuse features learned from separation for SELD. Instead, the proposed architecture reutilizes multiple auditory information estimated from common per-object features to refine the separation and SELD performance.

# 6 Conclusion

We proposed DeepASA for unified auditory scene analysis and separation, employing a multi-decoder architecture that utilizes relations between auditory information for both source separation and analysis tasks. The model leverages chain-of-inference to refine performance in USS, SED, and DoAE tasks. Experimental results demonstrated the effectiveness of sub-decoders and multi-clue attention in improving feature extraction. Our results show significant performance improvements over existing methods in various downstream datasets.

**Broader impacts** This paper focuses on the framework that emulates how humans perform ASA by utilizing various types of auditory information to segregate sound objects to solve the cocktail party problem. We believe that DeepASA offers a unified framework that not only advances the state of the art in auditory scene analysis and sound separation but also opens new directions for research at the intersection of spatial hearing, multi-task learning, and object-centric audio modeling.

**Limitations and future work** The main limitation is the large parameter size of the ATST adopted for the SED decoder. In this work, the separated object features are reprocessed by the patch-based ATST to keep the advantage of ATST pre-trained on a large set of audio classification datasets. For the aspect of the bias of the dataset, the reverberation time was set between 0.2 and 0.6 seconds, so the performance of the model may degrade in environments with reverberation times longer than 0.6 seconds. Additionally, because the background noise was simulated with an SNR range of 6 to 30 dB, the performance of the model is also expected to decline when the background noise has an SNR below 0 dB and is louder than the foreground source. Next, regarding the potential misuse, since this model can extract speech of individual speakers and analyze their types and directions, there is a concern that it could be exploited for eavesdropping.

Future research could explore pre-training the novel classification layer designed to be compatible with object features, using a large audio dataset like ATST.

## Acknowledgements and Disclosure of Funding

This work was supported by the National Research Foundation of Korea (NRF) grant (No. RS-2024-00337945) and STEAM research grant (No. RS-2024-00464269) funded by the Ministry of Science and ICT of Korea government (MSIT), the BK21 FOUR program through the NRF grant funded by the Ministry of Education of Korea government (MOE), and the Center for Applied Research in Artificial Intelligence (CARAI) funded by DAPA and ADD (UD230017TD).

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

# Appendix / Technical Appendices and Supplementary Material

This appendix is organized as follows:

- Appendix A provides detailed specifications for the SED decoder.
- Appendix B provides an in-depth analysis of the impact of the noise decoder on classification.
- Appendix C presents experiments on the effects of the direct and reverb decoders on DoAE.
- Appendix D examines the experimental results using the time-varying learnable window.
- Appendix E presents an analysis of the chain-of-inference mechanism.
- Appendix F compares the proposed method with SOTA models on the ASA2 dataset and includes a real-world demonstration.
- Appendix G presents the experimental results along with the number of sound sources.

## A   Detailed specifications of the SED decoder

In this section, we provide detailed specifications of the SED decoder. The SED decoder consists of three branches: ATST, T-CRNN, and F-CRNN. The specifications of the SED decoder are presented in Table 5. First, in the ATST branch, the object features are converted into patches of length $T_p = 4, F_p = 64$. The sequence length corresponds to the number of these transformed patches, and the patches are reshaped along the channel dimension. Since the pre-trained ATST takes a single-channel mel spectrogram as input, a linear layer is added before ATST to match the embedding dimension, followed by adaptive pooling to align the time frame to $T' = 40$. In the T-CRNN branch, seven convolution layers are used, as in [46], followed by a linear layer after concatenating the output of the ATST branch. The channel dimension and pooled frequency dimension are then reshaped into a single dimension, with the time dimension treated as the sequence dimension, and passed through a T-GRU. Finally, in the F-CRNN branch, pooling is focused on the time dimension, followed by two convolution layers, after which the channel dimension and pooled time dimension are reshaped into a single dimension, and the frequency dimension is treated as the sequence dimension before passing through an F-GRU. The combined output of the ATST and T-CRNN branches then passes through a fully connected layer with an output dimension of 1 to produce the object activation curve, a fully connected layer with an output dimension of $C = 13$ (the number of classes) to produce the sound event map, and is subsequently combined with the output of the F-CRNN after passing through attentive statistics pooling (ASP) [40] and a fully connected layer, producing the final class output.

Table 5: Detailed specifications of the SED decoder

| SED decoder | | |
|---|---|---|
| **ATST branch** | **T-CRNN branch** | **F-CRNN branch** |
| patchification $\frac{TF}{T_pF_p} \times DT_pF_p$ | (3,3)@64, LN, Mish | (3,3)@64, LN, Mish |
| | Pool 2×5 | |
| | (3,3)@64, LN, Mish | |
| | Pool 2×1 | Pool 4×5 |
| | (3,3)@64, LN, Mish | |
| FC 768 | Pool 2×1 | |
| Pre-trained ATST | (3,3)@64, LN, Mish | (3,3)@64, LN, Mish |
| | Pool 2×1 | |
| | (3,3)@64, LN, Mish | |
| AdaptivePool1D(250 → 40) | Pool 2×1 | Pool 4×4 |
| | (3,3)@64, LN, Mish | |
| FC320 | Pool 2×1 | |
| FC320 | | GRU 320 |
| GRU 320 | | |
| FC512, Dropout, FC1 | FC512, Dropout FC13 | Global pooling |
| | softmax | FC64, Tanh, FC13 / softmax / FC13 | FC512, Dropout, FC13 |
| **Activation** | **SED** | **Class** |

## B   Impact of the noise decoder on classification

We conducted a detailed analysis to investigate the impact of the noise decoder on classification. Figure 6 presents a class-wise comparison of classification recall performance between the cases

with and without the use of the noise decoder. Without the noise decoder, the classification recall for domestic sounds is 49.4%, which is lower than other classes. This is due to the similar time-frequency characteristics of domestic sounds and background noise (from the TAU-SNoise DB[2]), leading the model to misclassify domestic sounds as noise and remove them. However, after adding the noise decoder, the classification recall for domestic sounds increased significantly to 64.9%, showing an improvement of 15.5% points. Furthermore, the recall performance for almost classes improved. These results indicate that although the noise decoder estimates the noise waveform, it significantly influences classification performance.

In Figure 7, the t-SNE analysis further confirmed that without the noise decoder, domestic sounds (represented in purple) have unclear boundaries with other classes, but when the noise decoder is employed, the boundaries between the domestic sound and other classes are more clearly delineated. This suggests that the noise decoder explicitly estimates the noise, allowing the model to better distinguish between foreground sources and noise. In summary, employing a noise decoder addresses the issue in previous approaches where the undenoised background noise degraded classification performance.

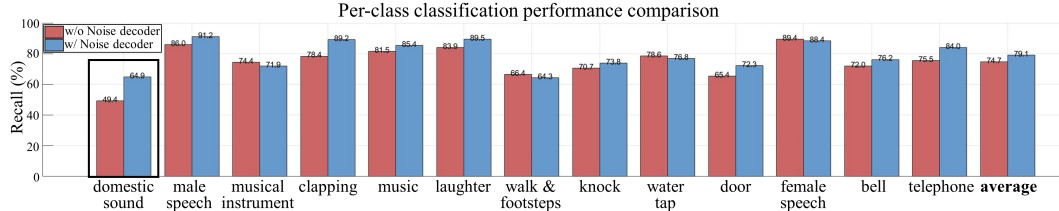

Figure 6: Per-class classification performance with and without noise decoder

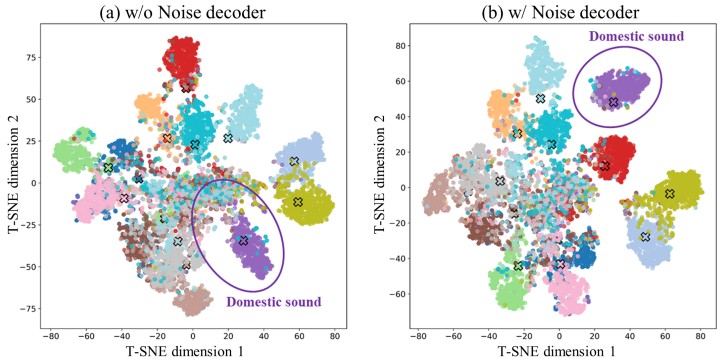

Figure 7: T-SNE comparison with and without noise decoder

## C  Effect of direct and reverb decoder on DoAE

To investigate the effect of estimating direct and reverberant foreground signals separately on DoAE, we conducted a detailed analysis. Estimating direct and reverberant signals separately introduces the dereverberation task in addition to the denoising and separation tasks, which can lead to a decrease in overall separation performance. However, as shown in Figure 8, the LE histogram for DoAE reveals an increase in the number of samples with LE within 10 degrees, while the number of samples with LE greater than 10 degrees decreases. This indicates that even though the direct and reverb decoders are part of the audio decoder, they positively influence DoAE performance.

To understand how the dereverberation task affects DoAE, we performed a cosine similarity analysis between the convolution kernel weights. Figure 9 compares the cosine similarity between the first convolution kernel weights of the DoA decoder and those of the direct and reverb decoders. The results show that the similarity between the DoA decoder and the direct decoder is higher than that between the DoA decoder and the reverb decoder. This suggests that the DoA decoder relies more on the direct components than the reverb components for DoAE. In reverberant environments, DoAE

---

[2]`https://zenodo.org/records/6408611`

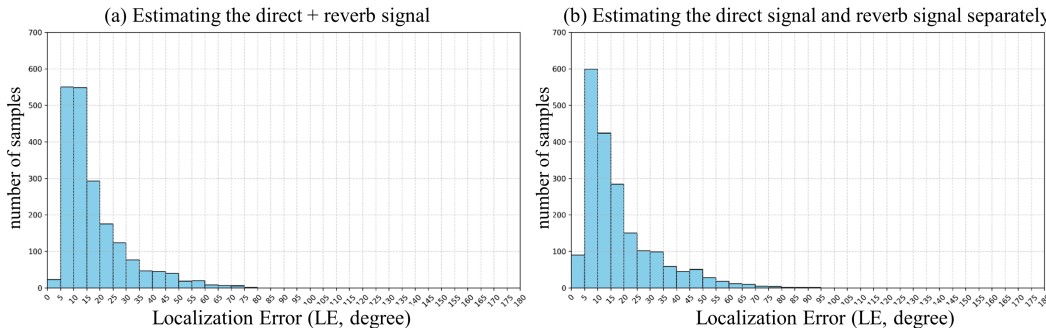

Figure 8: Comparison of LE histograms when estimating (a) direct + reverb signals together and (b) direct and reverb separately.

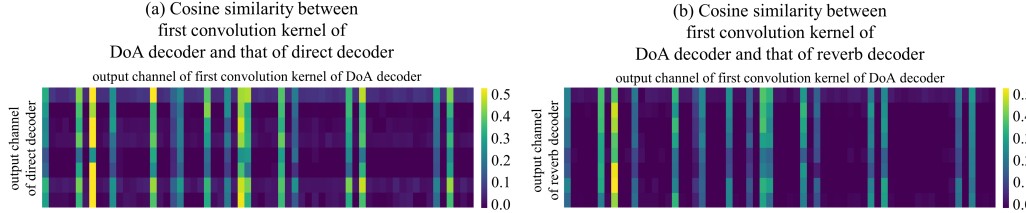

Figure 9: Comparison of cosine similarity between the weight of the first convolution kernel of the DoA decoder and the convolution kernel weights of (a) direct decoder and (b) reverb decoder

is more challenging compared to anechoic environments. This observation implies that learning to separate direct and reverb components within the object features can improve DoAE by ensuring that the direct components are properly embedded in the features.

Furthermore, to evaluate the performance of dereverberation, we compared the results of reverberant, direct (dereverberation, SI-SDR$_\mathbf{s}$) and reverb source separation (SI-SDR$_\mathbf{h}$), as well as the speech-to-reverberation modulation ratio (SRMR) [56]. SRMR is only calculated for classes corresponding to male and female speech. The experimental results demonstrate that the proposed model performs well in dereverberation, achieving performance comparable to full source separation. Additionally, with an SRMR of 6.5, the model indicates strong dereverberation effectiveness.

Table 6: Dereverberation performance of proposed model

| SI-SDR | SI-SDR$_\mathbf{s}$ | SI-SDR$_\mathbf{h}$ | SRMR |
|--------|---------|---------|------|
| 10.8 | 10.8 | 10.1 | 6.5 |

## D Detailed analysis on time-varying learnable window

In this section, we present experiments to investigate the role and functionality of the time-varying learnable window. In the audio encoder, the short-time Fourier transform (STFT) is widely used to convert a waveform into a complex spectrogram. Unlike speech separation, which considers only the speech class, USS involves various classes, necessitating dynamic adjustments to the configuration of the window. Although many conventional methods rely on fixed windows [28, 12, 38], this reliance limits their performance [57]. To address this limitation, we propose a time-varying learnable window that can dynamically adjust the window configuration for both USS and ASA.

We conducted a detailed ablation study on the effect of the time-varying learnable window. The results are presented in Table 7. Initially, we compared encoding methods using complex STFT and learnable 1D convolution kernels. The performance of complex STFT generally exceeded that of the 1D convolution kernel, suggesting that while the 1D convolution kernel extracts learnable features, complex STFT is more effective at explicitly capturing frequency information in the feature representation. Additionally, the time-invariant learnable window learns a fixed window across time, resulting in a window length that is similar to that of the STFT. The time-invariant learnable window can be obtained by performing mean pooling in the temporal dimension before being passed through the linear layers of the time-varying learnable window. Next, we examined TDY-CNN [57], which

Table 7: Ablation study results of time-variant learnable window

| Variation | USS | | SED | | DoAE | | SELD ↓ | Complexities | |
|---|---|---|---|---|---|---|---|---|---|
| | SI-SDRi ↑ | SDRi ↑ | ER ↓ | F1 ↑ | LE ↓ | LR ↑ | | Param. | MAC/s |
| STFT | 10.8 | 11.5 | 30.0 | 69.8 | 19.3 | 76.0 | 0.237 | **8.1 (+96.8) M** | **99.0 G** |
| 1D Conv | 7.8 | 8.8 | 40.2 | 59.0 | 27.1 | 61.2 | 0.338 | 9.0 (+96.8) M | 99.2 G |
| Time-invariant learnable window | 10.7 | 11.4 | 31.3 | 68.1 | 19.5 | 75.6 | 0.246 | 8.2 (+96.8) M | 99.1 G |
| TDY-CNN | 10.5 | 11.3 | 29.7 | 70.0 | 18.9 | 76.3 | 0.235 | 9.0 (+96.8) M | 99.4 G |
| **Time-variant learnable window** | **11.0** | **11.7** | **28.8** | **70.2** | **18.5** | **76.9** | **0.230** | 8.2 (+96.8) M | 99.1 G |

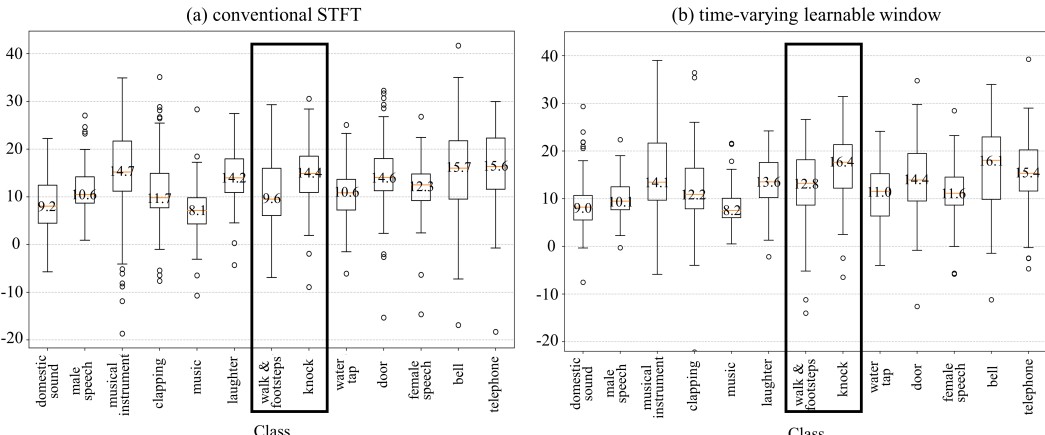

Figure 10: Comparison of source separation performance for each class: (a) conventional STFT, (b) time-varying learnable window

learns attention weights for each time frame to control how much information is passed through, capturing phoneme variation. However, this approach has the limitation of being unable to adjust the frequency band information. In contrast, the proposed method allows for control over the frequency band by adjusting the width of the main lobe.

Figure 10 presents a comparison of separation performance for each class. The results show a significant improvement in the separation performance of sporadic sounds, such as door and knock events. Despite being transient, the time-varying learnable window effectively captures the characteristics of these sound objects, allowing the model to improve separation performance.

Figure 11 shows a comparison of the sample between conventional STFT and the proposed time-varying learnable window. Figure 11(a) shows the time-domain waveform of the audio mixture, and Figure 11(b) plots its frame-wise standard deviation over time. Figure 11(c), (d), and (e) present the SED result of conventional STFT, that of time-varying learnable window, and ground truth, respectively, with binary values between 0 and 1 (black indicates 0, white indicates 1). The results show that with conventional STFT, object 1 was not captured, and object 3 failed in both classification

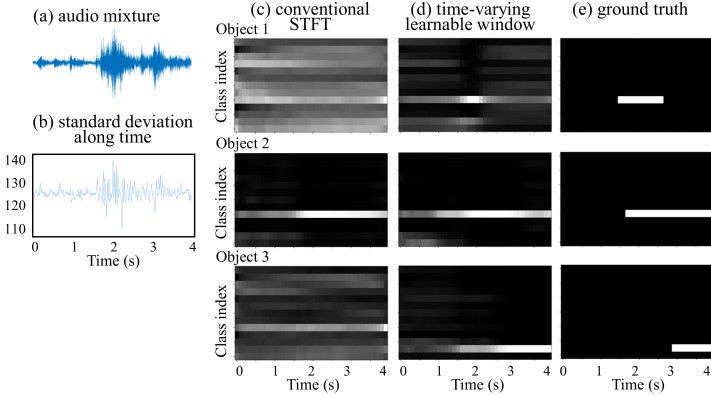

Figure 11: (a) Time-domain waveform of the audio mixture, (b) window length along the time frame, SED results for (c) conventional STFT, (d) time-varying learnable window, (e) ground truth.

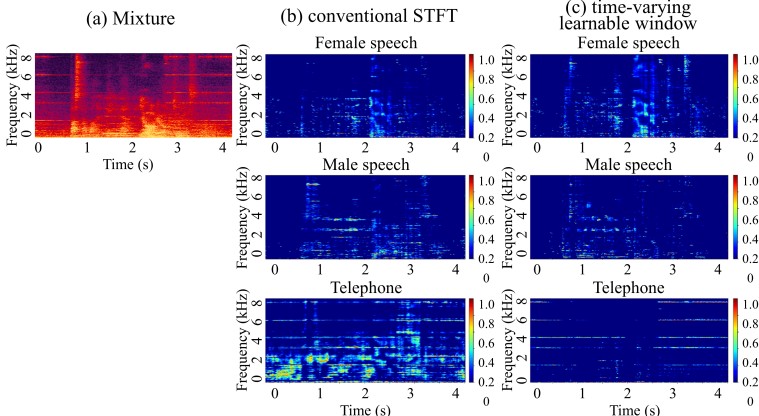

Figure 12: (a) Spectrogram of the audio mixture, GradCAM results for (b) conventional STFT, (c) time-varying learnable window.

and activation estimation. However, using the proposed method, the model can detect object 1, which improves the sound event detection performance of object 3.

Finally, Figure 12 shows a GradCAM [58] analysis of the separated object features immediately after the object splitter, to examine how well the object features encapsulate the properties of the corresponding sound objects. The model with conventional STFT struggled with object separation, as regions of speech are still embedded within telephone regions. In contrast, the proposed method more effectively isolates the telephone regions by suppressing most of the speech-related components. This demonstrates that the proposed approach effectively separates sound objects based on auditory information.

# E    Detailed study on Chain-of-Inference

In this section, we conduct a case study to understand the underlying mechanism of the CoI. It was observed that the model struggled with SED before applying the CoI, whereas the SED performance improved after its application. We then analyzed the fundamental reasons for this improvement. Figure 13(a), (b), and (c) present the SED output for the model without CoI, the model with CoI, and the ground truth, respectively. By comparing the performance before and after applying CoI, an improvement is observed in the SED estimation for object 1 and object 3. To investigate the cause of this difference, we examined the DoAE results for object 1 and object 3 before applying CoI in Figure 13(d) and (e), and the multi-clue attention maps with SED as the query and DoA as the query in Figure 13(f) and (g), respectively. In the DoAE results, for object 1, DoA vectors below the onset threshold (0.5) suggest the potential presence of an object, while for object 3, the estimated DoA closely matches the ground truth. This indicates that the classification of object 3 was refined through CoI by leveraging the nearly accurate DoAE. Furthermore, examining the attention maps, the attention map with SED as the query identified four objects, whereas the one with DoA as the query identified five. This demonstrates the successful correction of object 1, which was mistakenly classified as silence, by utilizing DoA information.

Table 8: Performance comparison by repeating chain-of-inference

| Stage | USS | | SED | | DoAE | | SELD ↓ | Complexities | |
|---|---|---|---|---|---|---|---|---|---|
| | SI-SDRi ↑ | SDRi ↑ | ER ↓ | F1 ↑ | LE ↓ | LR ↑ | | Param. | MAC/s |
| without CoI | 11.0 | 11.7 | 28.8 | 70.2 | 18.5 | 76.9 | 0.230 | 8.2 (+96.8) M | 99.1 G |
| **CoI (1st stage)** | **11.2** | **12.0** | **25.0** | **74.1** | **17.0** | **78.1** | **0.206** | 12.1 (+96.8) M | 104.0 G |
| CoI (2nd stage) | 11.1 | 11.9 | 26.8 | 72.3 | 17.2 | 77.5 | 0.216 | 16.0 (+96.8 M) | 118.9 G |

Additionally, the CoI mechanism utilizes the high-level information (DoA, SED) to refine the object feature separation based on the temporal coherence matching (TCM), so using the same mechanism multiple times can bias the results towards the direction of prioritizing TCM. We present the results of performing the chain-of-inference multiple times as Table 8. As a result, using CoI only once resulted in the highest performance.

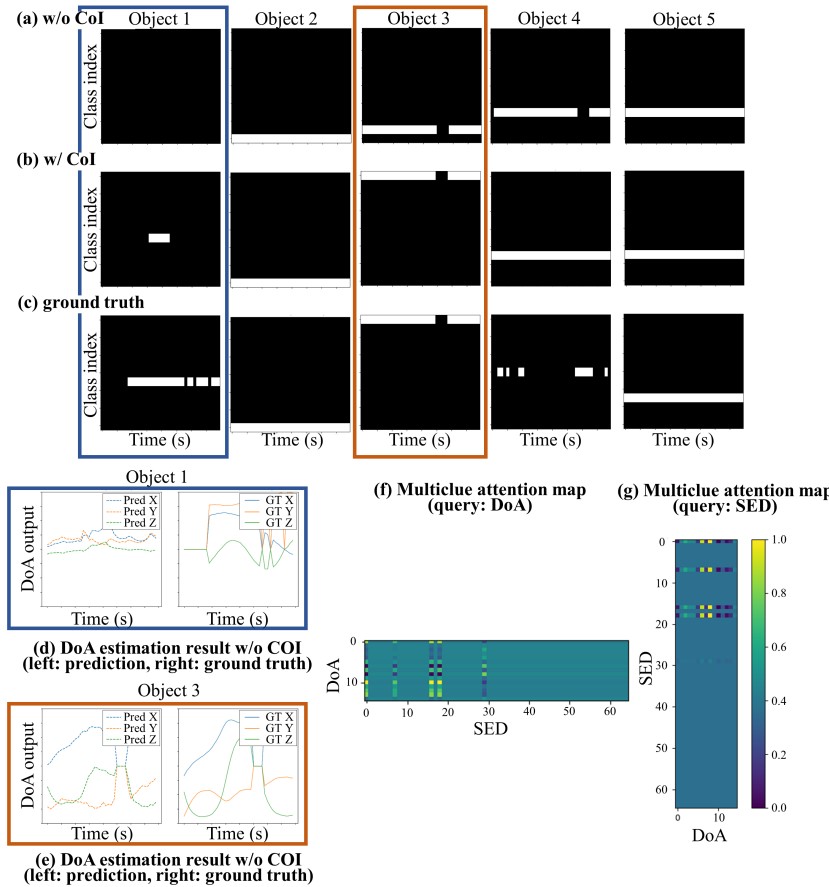

Figure 13: SED result (a) without CoI, (b) with CoI, (c) ground truth, and (d) DoAE result of object 1 without CoI model, (e) DoAE of object 3 without CoI, multiclue attention map for (f) DoA query, (g) SED query

## F  Comparison with SOTA models for ASA2 dataset and real-world demonstration

**Comparison USS and SELD performance on ASA2 dataset** We compared the performance of the proposed DeepASA model on the ASA2 dataset with that of other SOTA models in Table 9. First, the proposed model achieved superior USS performance compared to other models [26, 47, 49, 38], even though it operates with a MIMO setup and performs a dereverberation task. Despite having a lightweight feature aggregation module compared to DeFT-Mamba, the model still achieved SOTA performance. This improvement is likely due to the ability of the model to utilize various auditory information, such as class, activation, DoA, and noise, enabling better separation of foreground sources. Next, in terms of SELD performance, the proposed model significantly outperformed existing models. This is because ASA2 consists entirely of polyphonic audio, and earlier approaches that estimate the outputs directly from mixtures are unsuitable for such overlapped sources. Our model avoids this limitation by separating the objects.

**Real-world demonstration** To evaluate the applicability of the proposed method and dataset in real-world scenarios, we experimented using a pre-trained model without fine-tuning. The experiment was carried out in a real office environment, where three sound objects (male speech, music, and a telephone) were placed. Among these, the male speech was a moving source. The miniDSP ambiMIK-1 was utilized for capturing the sound, which is different from the microphone configuration for the pre-training dataset. The results demonstrate that the pre-trained DeepASA model can be applied to USS, SED, and DoAE tasks in real-world conditions, confirming its feasibility. A demo video of the

Table 9: Comparison with SOTA models on ASA2 dataset

| model | USS | | SED | | DoAE | | SELD ↓ | Complexities | |
|---|---|---|---|---|---|---|---|---|---|
| | SI-SDRi ↑ | SDRi ↑ | ER ↓ | F1 ↑ | LE ↓ | LR ↑ | | Param. | MAC/s |
| ByteDance-uss [7] | 5.2 | 8.3 | - | - | - | - | - | 28 (+80.7) M | 40.1 G |
| TF-GridNet [47] | 8.7 | 10.7 | - | - | - | - | - | 14.7 M | 462 G |
| SpatialNet [49] | 9.6 | 10.2 | - | - | - | - | - | 7.3 M | 71.8 G |
| DeFT-Mamba [38] | 10.4 | 11.3 | - | - | - | - | - | 3.6 M | 83.8 G |
| EINV2 [53] | - | - | 48.5 | 39.5 | 27.1 | 51.5 | 0.431 | 51.5 M | 6.7 G |
| ResNet Conformer [59] | - | - | 45.7 | 41.0 | 26.6 | 53.7 | 0.414 | 13.6 M | 7.6 G |
| SELD-Mamba [55] | - | - | 43.5 | 42.7 | 25.5 | 56.7 | 0.396 | 75.1 M | 4.3 G |
| MFF-EINV2 [50] | - | - | 42.1 | 43.2 | 25.8 | 60.7 | 0.381 | 54.8 M | 14.3 G |
| DeepASA (w/o ATST) | 11.0 | 11.7 | 30.1 | 69.5 | 18.5 | 74.8 | 0.240 | 8.2 M | 91.0 G |
| **DeepASA** | 11.0 | 11.7 | 28.8 | 70.2 | 18.5 | 76.9 | 0.230 | 8.2 (+96.8) M | 99.1 G |
| **(+) Chain-of-inference** | **11.2**±0.1 | **12.0**±0.1 | **25.0**±0.4 | **74.1**±0.3 | **17.0**±0.3 | **78.1**±0.4 | **0.206**±0.001 | 12.1 (+96.8) M | 104.0 G |

real-world experiment and example results from the ASA2 dataset are available on the page linked below[3].

Additionally, we conducted an experiment by fine-tuning the pretrained DeepASA over datasets generated from real-world room impulse responses (RIRs). The fine-tuned model was then evaluated by unseen real-world RIRs. From Table 10, the experimental results show a slight decrease in USS performance, as well as in most of SELD performance. This performance degradation can be attributed to the discrepancy in the hidden representation of simulation and real RIRs. Ideally, training with a large set of RIRs, including both simulated and measured RIRs, would improve the generalizability of models.

Table 10: Comparison between datasets with simulated and real-world RIRs

| RIR | USS | | SED | | DoAE | | SELD ↓ |
|---|---|---|---|---|---|---|---|
| | SI-SDRi ↑ | SDRi ↑ | ER ↓ | F1 ↑ | LE ↓ | LR ↑ | |
| simulated RIR | **11.0** | **11.7** | **28.8** | **70.2** | **18.5** | **76.9** | **0.230** |
| real RIR | 10.6 | 11.4 | 32.7 | 66.4 | 13.4 | 72.3 | 0.254 |

# G   Experimental results along with the number of sound sources

We performed inference when the number of sound sources was within the maximum estimable range [2, 5]. and when it exceeded the maximum estimable range [6, 7]. The experimental results are presented in Table 11. When the number of foreground sources is 6 or 7, resulting in a sharp decline in performance. The experimental results show that when there are 6 sources, two sources are not separated, leading to very low performance only for the two sources. When there are 7 sources, three sources are mixed and mapped to one track in most cases, resulting in very low performance for three of the sources.

Table 11: Experimental results along with the number of sound sources

| Number of sources | | USS | | SED | | DoAE | | SELD ↓ |
|---|---|---|---|---|---|---|---|---|
| | | SI-SDRi ↑ | SDRi ↑ | ER ↓ | F1 ↑ | LE ↓ | LR ↑ | |
| 2 | | **17.7** | **17.1** | **22.1** | **82.4** | **10.5** | **88.5** | **0.143** |
| 3 | | 14.6 | 14.5 | 22.4 | 80.8 | 13.1 | 86.3 | 0.160 |
| 4 | | 8.5 | 10.6 | 33.8 | 65.6 | 20.9 | 72.5 | 0.269 |
| 5 | | 7.9 | 10.3 | 34.9 | 64.5 | 22.1 | 71.3 | 0.280 |
| 6 | Total | 2.4 | 5.9 | 47.7 | 48.5 | 40.3 | 54.5 | 0.418 |
| | Top 4 average | 7.0 | 9.6 | 38.2 | 61.2 | 25.6 | 66.7 | 0.311 |
| | Bottom 2 average | -6.8 | -1.5 | 69.1 | 23.1 | 69.7 | 30.1 | 0.636 |
| 7 | Total | 0.2 | 4.5 | 59.0 | 36.6 | 51.7 | 42.4 | 0.521 |
| | Top 4 average | 6.6 | 9.1 | 38.9 | 60.6 | 29.2 | 64.5 | 0.325 |
| | Bottom 2 average | -8.3 | -1.6 | 85.8 | 4.6 | 81.7 | 12.9 | 0.784 |

---

[3] https://huggingface.co/spaces/donghoney22/DeepASA

