# OpenReview forum: "DeepASA: An Object-Oriented Multi-Purpose Network for Auditory Scene Analysis"
_NeurIPS.cc/2025/Conference — NeurIPS 2025 poster_

### Official Review · Reviewer_Mddv · 2025-06-24

**Clarity:** 2
**Significance:** 3
**Originality:** 2
**Rating:** 4
**Confidence:** 4

**Summary:**

The paper proposes DeepASA, a new model for auditory scene analysis, taking in a multichannel audio mixture capable of performing source separation, sount event detection, audio classification and direction-of-arrival estimation. The DeepASA model itself is based on the DeFT-Mamba model.
The two core contributions are the chain-of-inference architecture and object oriented processing.

**Questions:**

1. In lines 307-308 you write
>This performance gain is largely attributed to the noise decoder’s explicit estimation of background noise and the benefits of fine-tuning from pre-trained weights.

This is something I've wondered about when I saw Figure 2; why exactly is background noise estimated and how would it contribute to an improvement in estimating the foreground sources?

2. Figure 2 shows how the channel dimension is increased from $2M$ to $D$ but the text (line 114) claims it increases from $2M$ to $C$. Is that just a mistake, given that $C$ is later (line 162) defined as the number of classes?

**Ethical Concerns:**

["NO or VERY MINOR ethics concerns only"]

**Final Justification:**

The provided updated experiments, particularly regarding the computational cost of the proposed architecture made me increase the rating to borderline accept.

**Limitations:**

Yes

**Paper Formatting Concerns:**

This is a minor nitpick, but many metrics are only named by their abbreviations (SI-SDRi, SDRi, SA-SDR) and should be defined the first time they are named, e.g. scale-invariant signal-to-distortion ratio improvement (SI-SDRi), etc.

The same is true for $D$, $T'$ and $C$ in Figure 2. It is not immediately obvious what they are supposed to be from the Figure alone.

The description of Table 3 should also mention it is the SI-SDRi in dB which is reported.

**Quality:**

3

**Strengths And Weaknesses:**

Strengths
- DeepASA is capable of performing universal sound separation, sound event detection and direction-of-arrival estimation all within one model
- DeepASA outperforms previous models in most metrics on the three datasets it was tested on
- The proposed model is significantly more accurate with more than two foreground sources compared to the previous work on the MC-FUSS dataset
- Detailed ablation study was done for all the proposed changes


Weaknesses
- DeepASA has significantly higher computational cost than DeFT-Mamba, the model it is based on despite often only achieving modest accuracy gains.
- Chain-of-inference is defined as one of the two major contributions of the paper. But the results do not seem to warrant this focus. In fact, the biggest improvements over previous models can be seen in the MC-FUSS dataset where chain-of-inference is not even used at all.
- Other than the MC-FUSS, the accuracy improvements of the DeepASA over previous work are more modest, a combination of many minor improvements adding up.

---

> ### Author Rebuttal · Authors · 2025-07-31
>
> **Weak-Q1.** DeepASA has significantly higher computational cost than DeFT-Mamba, the model it is based on despite often only achieving modest accuracy gains.
>
> **A.** (1) Compared to DeFT-Mamba, DeepASA covers a wider range of tasks. It can perform separation, denoising, dereverberation, sound event detection, and DoA estimation for up to 5 sources. In the ASA2 dataset, it showed an improvement of 0.8 dB in SI-SDR and 0.7 dB in SDR compared to DeFT-Mamba.
>
> (2) When only performing USS, DeepASA has a lower computational load but higher performance than DeFT-Mamba. Initially, we incorrectly reported the computational cost on the MC-FUSS dataset. In the MC-FUSS dataset, only the audio decoder is used, and the other decoders are not utilized, which reduces the parameter size and computational complexity of DeepASA.
> To ensure a fair comparison with DeFT-Mamba, we compared the performance of DeepASA trained from scratch on the MC-FUSS dataset without using the pre-trained ATST with the performance in Table 3. The experimental results are as follows: DeepASA without using a pre-trained ATST has a lower parameter size and computational complexity than DeFT-Mamba, but its performace improved for all source counts, even when it is trained from scratch. This is attributed to the exclusion of F-Mamba, which improves sound separation performance but has a high computational load, and the introduction of the proposed dynamic STFT and noise decoder. We will add these experimental results to the main paper.
>
> ASA2 comparison between DeepASA with and without ATST
> |Model|SI-SDRi|SDRi|ER|F1|LE|RL|SELD|Param.|MAC/s|
> |-|-|-|-|-|-|-|-|-|-|
> |DeFT-Mamba|10.4|11.3||||||**3.6 M**|**83.8 G**|
> |DeepASA (w/o ATST)|11.0|11.7|30.1|69.5|18.5|74.8|0.240|8.2 M|91.0 G|
> |DeepASA (w/ ATST)|11.0|11.7|28.8|70.2|18.5|76.9|0.230|105.0 M|99.1 G|
> |DeepASA (w/ ATST, w/ CoI)|**11.2**|**12.0**|**25.0**|**74.1**|**17.0**|**78.1**|**0.206**|108.9 M|104.0 G|
>
> MC-FUSS comparison between DeepASA with and without ATST (SI-SDRi)
> (S denotes the number of sources in the mixture).
> |Model|S=2|S=3|S=4|Total|Param.|MAC/s|Runtime|
> |-|-|-|-|-|-|-|-|
> |DeFT-Mamba|18.4|17.1|13.8|16.4|3.6 M|83.8 G|39.2 min|
> |DeepASA (w/o ATST, w/o pre-train)|18.5|18.2|15.7|17.5|**2.9 M**|**77.3 G**|**26.8 min**|
> |DeepASA (w/ ATST, w/ pre-train)|**18.9**|**19.1**|**17.6**|**18.5**|**2.9 M**|**77.3 G**|**26.8 min**|
>
> **Weak-Q2.** Chain-of-inference is defined as one of the two major contributions of the paper. But the results do not seem to warrant this focus. In fact, the biggest improvements over previous models can be seen in the MC-FUSS dataset where chain-of-inference is not even used at all.
>
> **A.** The improvement on the MC-FUSS dataset is due to the comprehensive object separation and analysis introduced in deepASA. These include dereverberation, multichannel-to-multichannel (M2M) separation, dynamic STFT process, and noise estimation. These modifications brought the biggest improvements when only the USS task was performed. However, the improvement on the USS task does not mean the inefficacy of the chain-of-inference (CoI). In our ablation study with and without CoI, the model with CoI demonstrates +0.3 dB in the SDRi and -0.024 in a SELD score, compared to the model without CoI in the ASA2 dataset. Additionally, the model is able to surpass the existing models by using CoI in the STARSS23 dataset.
>
> ASA2 dataset results compared with and without chain-of-inference and state-of-the-art models
> ||SI-SDRi↑|SDRi↑|ER↓|F1↑|LE↓|LR↑|SELD↓|
> |-|-|-|-|-|-|-|-|
> |DeFT-Mamba|10.4|11.3||||||
> |MFF-EINV2|||42.1|43.2|25.8|60.7|0.381|
> |DeepASA(w/oCoI)|11.0|11.7|28.8|70.2|18.5|76.9|0.230|
> |DeepASA(w/CoI)|**11.2**|**12.0**|**25.0**|**74.1**|**17.0**|**78.1**|**0.206**|
>
> Additionally, we applied the CoI to existing SELD models and observed performance improvements. Existing models do not employ object-oriented processing, so only the cross-attention from temporal coherence matching was applied to the existing models. The experimental results are shown in the table below, and since both models exhibited similar enhancements, this demonstrates the generalizability of CoI.
>
> Influence of employing CoI to existing SELD models
> ||Col|ER↓|F1↑|LE↓|LR↑|SELD↓|
> |-|-|-|-|-|-|-|
> |MFF-EINV2|X|42.1|43.1|25.8|60.7|0.381|
> ||O|**38.7**|**47.2**|**22.1**|**63.5**|**0.351**|
> |SELD-Mamba|X|43.5|42.7|25.5|56.7|0.396|
> ||O|**40.1**|**46.6**|**22.7**|**61.1**|**0.363**|
>
> **Weak-Q3.** Other than the MC-FUSS, the accuracy improvements of the DeepASA over previous work are more modest, a combination of many minor improvements adding up.
>
> **A.**
> - In terms of performance, the benchmark for the STARSS23 dataset in Table 4 of the DCASE challenge permitted the use of pre-trained models and external large datasets, as well as data augmentation, resulting in each system using different training data and pre-trained models. Consequently, in Table 8 of the Appendix, the performance of SOTA models for the sound event localization and detection (SELD) task without the use of external data or data augmentation is listed. The comparison with the performance of DeepASA without using the pre-trained ATST can be found in the reply to the previous comment (Weak-Q1-A). Ultimately, DeepASA achieved a 0.6 dB improvement over the best performance of previous works in terms of SI-SDRi and a 0.141 improvement in SELD score compared to the best performance of previous works in a fair comparison. This can be considered a sufficiently significant gap.
> - Also, the main contribution of DeepASA is not its exceptionally high performance on a specific task but rather its ability to enhance performance across various tasks by learning multiple auditory information (class, timestamp, and DoA) through several sub-decoders to separate object features. In other words, DeepASA contributes not only in terms of model architecture but also in its training methodology. From a model structure perspective, it improves efficiency by excluding the computationally intensive F-Mamba and allows for the learning of time resolution and frequency lobes over time through dynamic STFT. Additionally, it enhances the estimation performance of individual auditory information through cross-referencing multiple auditory information sources, similar to human processing. The DeepASA and chain-of-inference structure is a novel architecture not previously proposed.
>
> **Question-Q1.** In lines 307-308 you write "This performance gain is largely attributed to the noise decoder’s explicit estimation of background noise and the benefits of fine-tuning from pre-trained weights."
>
> **A.** Estimating background noise is essential because, when only the foreground source is separated, noise often mixes into the inactive source, which should be estimated as silence. To prevent this, noise separation is explicitly learned. A similar approach based on loss functions has existed in speech enhancement [**]. Specifically, by estimating noise, the sum of the foreground source and noise is constrained to be the audio mixture, which indirectly constrains the phase of the estimated foreground sources. This helps reduce artifacts that occur when using loss in the STFT domain. This approach also contributes to improving classification performance in this paper.
>
> [**] A. Pandey, D.L. Wang, "Dense CNN with self-attention for time-domain speech enhancement." IEEE/ACM Trans. Audio, Speech, Lang. Process., 29, pp. 1270 – 1279, 2021.
>
> **Question-Q2.** Figure 2 shows how the channel dimension is increased from 2M to D but the text (line 114) claims it increases from 2M to C. Is that just a mistake, given that C is later (line 162) defined as the number of classes?
>
> **A.** Thank you for the reviewer’s comment. We defined C as the number of classes, and it needs to be D. We will change “from 2M to C” into “from 2M to D.”
>
> **Paper Formatting Concerns.** This is a minor nitpick, but many metrics are only named by their abbreviations (SI-SDRi, SDRi, SA-SDR) and should be defined the first time they are named, e.g. scale-invariant signal-to-distortion ratio improvement (SI-SDRi), etc. The same is true for D, T' and C Figure 2. It is not immediately obvious what they are supposed to be from the Figure alone. The description of Table 3 should also mention it is the SI-SDRi in dB which is reported.
>
> **A.** Thank you for pointing that out. We will define the first time the evaluation metrics are named. (scale-invariant signal-to-distortion ratio improvement (SI-SDRi), signal-to-distortion ratio (SDR), source-aggregated signal-to-distortion ratio (SA-SDR). As the reviewer commented, we will define D, T’, and C in the caption of Figure 2. We will mention that SI-SDRi is reported on the dB scale.

---

> > ### Comment · Reviewer_Mddv · 2025-08-09
> >
> > Thank you for providing these additional results. The updated computational cost comparison between the proposed architecture and the previous best in particular are very convincing.
> > I will increase the paper score accordingly to borderline accept.

---

### Official Review · Reviewer_Qqmd · 2025-07-02

**Clarity:** 2
**Significance:** 3
**Originality:** 3
**Rating:** 4
**Confidence:** 4

**Summary:**

This paper presents DeepASA, a unified framework for auditory scene analysis that performs multichannel source separation, dereverberation, sound event detection, audio classification, and direction-of-arrival estimation in one model. The core contributions are: (1) an Object-Oriented Processing (OOP) strategy, which isolates per-object audio features early in the pipeline via a transformer-based aggregator and object separator, and (2) a Chain-of-Inference (CoI) mechanism that iteratively refines object features by fusing sub-decoder outputs through temporal-coherence attention. Evaluated on three benchmarks (ASA2, MC-FUSS, STARSS23), DeepASA achieves state-of-the-art performance across all tasks.

**Questions:**

See Weaknesses.

**Ethical Concerns:**

["NO or VERY MINOR ethics concerns only"]

**Final Justification:**

The paper contributes a unified framework for auditory scene analysis. The experiments demonstrate its effectiveness. After reading other reviewers' comments, I would like to recommend Borderline Accept.

**Limitations:**

The authors discuss the ATST size but could further elaborate on dataset biases (e.g., reverberation conditions) and potential misuses (e.g., surveillance).

**Quality:**

3

**Strengths And Weaknesses:**

Strengths:
1. Ablations isolate each module’s contribution; end-to-end performance on multiple datasets is compelling.
2. Real-world examples on the demo webpage lend practical credibility.
3. Well-designed multi-decoder architecture eliminates permutation issues and enables complementary cue integration.

Weaknesses:
1. Figures 2 & 5 (model diagram & CoI) are dense and lack intuitive legends. Readers may struggle to follow data flow. I think a step-by-step schematic or callouts would help.
2. Dynamic STFT (Sec. 2.2) is novel but under-explained: it would be clearer with visualizations of learned windows over time and ablations on σₜ initialization.
3. The choice of maximum object count J is not justified: how sensitive is performance if fewer/more sources appear?
4. Computational cost of the ATST-based SED decoder is high. It would be great to discuss the runtime and efficiency trade-offs.
5. Related Work (Sec. 5) omits object-centric learning: key works like "Slot Attention" (Locatello et al., 2020), "Sound of Pixels" (Zhao et al., 2018), "Objects That Sound" (Arandjelović et al., 2017), "SSV2A" (Guo et al., 2024), "AVObject" (Li et al., 2025), are not discussed. The distinction between those and DeepASA’s OOP should be clarified.

---

> ### Author Rebuttal · Authors · 2025-07-31
>
> **Weak-Q1.** Figures 2 & 5 (model diagram & CoI) are dense and lack intuitive legends. Readers may struggle to follow data flow. I think a step-by-step schematic or callouts would help.
>
> **A.** Thank you for pointing this out. We will revise the figures by creating a step-by-step schematic and adding detailed explanations in the captions so that the model structure can be understood just by looking at the figures. Specifically, in Figure 2, we aim to highlight how each object is processed in parallel on a batch basis through the object splitter. In Figure 5, we will clarify the signal flow from the SED and DoA query branches to TCM in the chain-of-inference.
>
>
> **Weak-Q2.** Dynamic STFT (Sec. 2.2) is novel but under-explained: it would be clearer with visualizations of learned windows over time and ablations on σₜ initialization.
>
> **A.** Thank you for mentioning the dynamic STFT. We briefly introduced its ablation study and the visualization of the learned window (changes in standard deviation over time) in Appendix D, but as per the reviewer's suggestion, we will provide a more detailed visualization in the revised paper. First, we will add visualizations of the learned window (mean and standard deviation) from various samples. To summarize the results here, the mean slightly alters around zero for most samples, but the standard deviation (width of window) dynamically changes, indicating that the dynamic STFT is utilized for adjusting the temporal and spectral resolution.
> - The ablation study on initialization of $\sigma_t$ shows that proper initialization is important. If $\sigma_t$ is not initialized with a size comparable to the width of the STFT analysis window, e.g., zero initialization, the features after the encoder do not sufficiently contain the information of the audio mixture, and the subsequent parameters cannot be trained for each task. Therefore, initialization is an essential element for learning, and as mentioned in the main text, $\sigma_t$ was set to 6.67 ms to ensure that $6\sigma_t$ can cover one frame.
>
>
> **Weak-Q3.** The choice of maximum object count J is not justified: how sensitive is performance if fewer/more sources appear?
>
> **A.** We set J=5 because the minimum maximum number of sources that allows the pre-trained model to be fine-tuned on the STARSS23 dataset is 5.
>  We performed inference when the number of sound sources is within the maximum estimable range (S=2–5<=J) and when it exceeded the maximum estimable range (S=6–7>J). When the number of foreground sources is 6 or 7, resulting in a sharp decline in performance. The experimental results show that when there are 6 sources, two sources are not separated, leading to very low performance only for the two sources. When there are 7 sources, three sources are mixed and mapped to one track in most cases, resulting in very low performance for three of the sources.
>
>
> ASA2 dataset result for changing the number of sources (S)
> ||SI-SDRi↑|SDRi↑|ER|F1↑|LE↓|LR↑|SELD↓|
> |-|-|-|-|-|-|-|-|
> |S=2|**17.7**|**17.1**|**22.1**|**82.4**|**10.5**|**88.5**|**0.143**|
> |S=3|14.6|14.5|22.4|80.8|13.1|86.3|0.160|
> |S=4|8.5|10.6|33.8|65.6|20.9|72.5|0.269|
> |S=5|7.9|10.3|34.9|64.5|22.1|71.3|0.280|
> |S=6|2.4|5.9|47.7|48.5|40.3|54.5|0.418|
> |S=7|0.2|4.5|59.0|36.6|51.7|42.4|0.521|
>
> Detailed results for S=6 and 7
> |||SI-SDRi↑|SDRi↑|ER↓|F1↑|LE↓|LR↑|SELD↓|
> |-|-|-|-|-|-|-|-|-|
> |S=6|Top 4 average|**7.0**|**9.6**|**38.2**|**61.2**|**25.6**|**66.7**|**0.311**|
> ||Bottom 2 average|-6.8|-1.5|69.1|23.1|69.7|30.1|0.636|
> |S=7|Top 4 average|**6.6**|**9.1**|**38.9**|**60.6**|**29.2**|**64.5**|**0.325**|
> ||Bottom 2 average|-8.3|-1.6|85.8|4.6|81.7|12.9|0.784|
>
>
> **Weak-Q4.** Computational cost of the ATST-based SED decoder is high. It would be great to discuss the runtime and efficiency trade-offs.
>
> **A.** We additionally compared the performance of the final DeepASA model without the pre-trained ATST on the ASA2 dataset and the MC-FUSS dataset. The experimental results showed that while the parameter size significantly decreased, the model's performance did not change significantly, and the reduced performance still surpassed other existing state-of-the-art models. Specifically, on the MC-FUSS dataset, DeepASA achieved higher performance than DeFT-Mamba, even without the pre-trained ATST and pre-training on the ASA2 dataset, despite its smaller parameter size, computational complexity, and runtime (training time per epoch). We will add these experimental results to the main text.
>
> ASA2 comparison between DeepASA with and without ATST
> |DeepASA|SI-SDRi|SDRi|ER|F1|LE|LR|SELD|Param.|Runtime|MAC/s|
> |-|-|-|-|-|-|-|-|-|-|-|
> |w/ ATST|**11.0**|**11.7**|**28.8**|**70.2**|**18.5**|**76.9**|**0.230**|105.0 M|30.1 min|99.1 G|
> |w/o ATST|**11.0**|**11.7**|30.1|69.5|**18.5**|74.8|0.240|**8.2 M**|**27.6 min**|**91.0 G**|
>
> MC-FUSS comparison between DeepASA with and without ATST (SI-SDRi)
> (S denotes the number of sources in the mixture).
> |Model|S=2|S=3|S=4|Total|Param.|MAC/s|Runtime|
> |-|-|-|-|-|-|-|-|
> |DeFT-Mamba|18.4|17.1|13.8|16.4|3.6 M|83.8 G|39.2 min|
> |DeepASA (w/o ATST, w/o pre-train)|18.5|18.2|15.7|17.5|**2.9 M**|**77.3 G**|**26.8 min**|
> |DeepASA (w/ ATST, w/ pre-train)|**18.9**|**19.1**|**17.6**|**18.5**|**2.9 M**|**77.3 G**|**26.8 min**|
>
>
> **Weak-Q5.** Related Work (Sec. 5) omits object-centric learning: key works like "Slot Attention" (Locatello et al., 2020), "Sound of Pixels" (Zhao et al., 2018), "Objects That Sound" (Arandjelović et al., 2017), "SSV2A" (Guo et al., 2024), "AVObject" (Li et al., 2025), are not discussed. The distinction between those and DeepASA’s OOP should be clarified.
>
> **A.** [Slot Attention] The key difference is that Slot Attention sets the slots first and then trains the mapping of objects to these slots, whereas OOP learns the spatial, frequency, and temporal relationships of the mixture audio first and then separates the objects through an object splitter.
>
>  The main difference between these works and the proposed OOP is that, unlike OOP’s core characteristic, these works do not process the separated features in parallel by permuting them along the batch dimension for each task (localization, separation, retrieval, and audio generation). In OOP, features are separated by the object splitter and processed independently in the batch dimension, fundamentally preventing interference from other object features.
>
>  [Sound of Pixels] is a network that separates audio occurring at specific pixels on a pixel-level basis. It extracts video features through a video analysis network and audio features through an audio analysis network and then performs source separation in an audio synthesis network. This approach is similar to conventional audio source separation networks, which extract information for source separation and ultimately perform the separation.
>
>  [Object that Sounds] proposes a network that fuses video and audio features for cross-modal retrieval, enabling semantic object localization. Similar to the previous Sound of Pixels, it utilizes object-centric learning. If OOP were applied to this work, features could be extracted through AVEnet, and the object splitter would separate each audio feature, allowing parallel retrieval of each sound source at the batch level.
>
>  [SSV2A, AVObject] also employ object-centric learning in the multi-modal domain, similar to the previous two papers. In conclusion, the key difference between object-centric learning and OOP lies in whether objects are explicitly separated and then processed in parallel through a parameter-shared decoder by passing them as batches.
>
>
> **Limitation-Q1.** The authors discuss the ATST size but could further elaborate on dataset biases (e.g., reverberation conditions) and potential misuses (e.g., surveillance).
>
> **A.** Thank you for the suggestion. We will add the following concerns in the Limitation & Future Work section: The reverberation time was set between 0.2 and 0.6 seconds, so the model’s performance may degrade in environments with reverberation times longer than 0.6 seconds. Additionally, because the background noise was simulated with an SNR range of 6 to 30 dB, the model’s performance is also expected to decline when the background noise has an SNR below 0 dB and is louder than the foreground source.
>
> Next, regarding the potential misuse, since this model can extract individual speakers’ speech and analyze their types and directions, there is a concern that it could be exploited for eavesdropping.

---

### Official Review · Reviewer_FCLK · 2025-07-02

**Clarity:** 3
**Significance:** 3
**Originality:** 3
**Rating:** 4
**Confidence:** 4

**Summary:**

Authors present DeepASA as a unified framework for acoustic scene analysis, adopting an Object-Oriented Processing approach that separates object-centric features at an early stage to avoid permutation ambiguity, and the Chain-of-Inference mechanism that iteratively refines predictions using temporal coherence matching between different auditory cues

**Questions:**

Could authors provide evidence for true object correspondence? .. some validation that object features actually correspond to individual sound sources rather than just being abstract encoded representations.

Is it possible to show that each of the J=5 object features consistently tracks the same sound source across time? When sounding objects enter and exit a scene, are object features able to maintain semantic coherence?

**Ethical Concerns:**

["NO or VERY MINOR ethics concerns only"]

**Limitations:**

yes.

**Paper Formatting Concerns:**

no.

**Quality:**

3

**Strengths And Weaknesses:**

Strengths
This paper addresses a fundamental challenge of comprehensive scene analysis, maintaining consistent object-wise associations across all sub-decoders task heads, as opposed to prior track0wise processing systems.

The motivation for object separator, CoI, as well as dynamic stft is clearly articulated and supported by ablation evidence. Temporal coherence matching is also shown to provide measurable improvements.

Weakness:
The overall model, even with pre-trained AST is notably bulky compared to task-specific approaches and is a concern for practical deployment. Additionally, given the multistage training, can authors provide an analysis of training convergence properties and sensitivity to initialization. Are there convergence guarantees for the chain-of-inference mechanism? I believe the 97M AST limits the novelty for SED component + is a computational bottleneck.

With regards to ASA2 dataset, the balanced distribution proportional to source count and increased maximum sources from 4 → 5 compared to prior versions suggests substantial modifications that could favor the proposed approach. Overall paper can benefit from highlighting some potential biases in ASA2

Foreground sources are fixed to J=5, however, some analysis regarding possible performance degradation when this assumption is violated would be helpful. Some analysis varying the degree of polyphonicity is expected. What is the computational and memory scaling behaviour?

Minor weakness:
The relationship between different loss components and their relative weighting could be better motivated in the main paper than in the appendix.

---

> ### Author Rebuttal · Authors · 2025-07-31
>
> **Weak-Q1.** The overall model, even with pre-trained AST is notably bulky compared to task-specific approaches and is a concern for practical deployment. Additionally, given the multistage training, can authors provide an analysis of training convergence properties and sensitivity to initialization. Are there convergence guarantees for the chain-of-inference mechanism? I believe the 97M AST limits the novelty for SED component + is a computational bottleneck.
>
> **A.** As for the concern on the 97M ATST, we conducted an extra experiment using the same training dataset (ASA2) without using pre-trained ATST. The experimental results indicate that the USS performance remains consistent, and the SELD score is slightly lowered by 0.01. When compared to other state-of-the-art models in Table 8 of Appendix F, the performance does not differ significantly. The ATST was used to achieve the best SELD performance possible, but it can be omitted when the space complexity matters. In addition, the size of ATST (97M) is much larger than DeepASA without ATST (8.2M) but similar to those of SELD-Mamba (75.1 M) and MFF-EINV2 (54.8M).
>
> Comparison between DeepASA models with and without ATST
> |DeepASA|SI-SDR↑|SDR↑|ER↓|F1↑|LE↓|LR↑|SELD↓|Param.|MAC/s|
> |-|-|-|-|-|-|-|-|-|-|
> |w/ pre-trained ATST|**11.0**|**11.7**|**28.8**|**70.2**|**18.5**|**76.9**|**0.230**|105.0 M|99.1 G|
> |w/o ATST|**11.0**|**11.7**|30.1|69.5|**18.5**|74.8|0.240|**8.2 M**|**91.0 G**|
>
> For the initialization issue, we conducted experiments 5 times with different initializations and included the mean and standard deviation in the last row of Table 2. The influence of different initializations was not significant, only showing std 0.001 in SELD score and 0.1 dB in SI-SDR.
>
> The CoI mechanism utilizes the high-level information (DoA, SED) to refine the object feature separation based on the temporal coherence matching (TCM), so using the same mechanism multiple times can bias the results towards the direction of prioritizing TCM. We present the results of performing the chain-of-inference multiple times as follows.
>
> Comparison of multi-stage COIs
> |DeepASA|SI-SDR↑|SDR↑|ER↓|F1↑|LE↓|LR↑|SELD↓|Param.|MAC/s|
> |-|-|-|-|-|-|-|-|-|-|
> |w/o CoI|11.0|11.7|28.8|70.2|18.5|76.9|0.230|**105.0 M**|**99.1 G**|
> |CoI (1 stage)|**11.2**|**12.0**|**25.0**|**74.1**|**17.0**|**78.1**|**0.206**|108.9 M|104.0 G|
> |CoI (2 stages)|11.1|11.9|26.8|72.3|17.2|77.5|0.216|112.8 M|118.9 G|
>
> **Weak-Q2.** With regards to ASA2 dataset, the balanced distribution proportional to source count and increased maximum sources from 4 → 5 compared to prior versions suggests substantial modifications that could favor the proposed approach. Overall paper can benefit from highlighting some potential biases in ASA2
>
> **A.** In Table 8, all models were trained using the same ASA2 dataset, so the composition of this dataset does not particularly favor the proposed approach. However, since the pre-trained ATST on AudioSet might have an impact, we will add the performance of DeepASA without using the pre-trained ATST to the table (refer to the previous reply).
>
>  As the reviewer mentioned, the pre-trained DeepASA could be advantageously applied for fine-tuning on the MC-FUSS dataset. To ensure transparency in performance evaluation, we present the performance of DeepASA trained from scratch on the MC-FUSS dataset, also without using a pre-trained model (ATST), similar to other models in Table 3.  DeepASA without pretraining and ATST achieves performance surpassing the previous best, DeFT-Mamba, with lower parameter size, computational complexity, and runtime (training time per epoch).
>
> MC-FUSS dataset result comparison with DeepASA trained from scratch (w/o ATST)
> (S denotes the number of sources in the mixture).
> ||S=2|S=3|S=4|Total|Param.|MAC/s|Runtime|
> |-|-|-|-|-|-|-|-|
> |DeFT-Mamba|18.4|17.1|13.8|16.4|3.6 M|83.8 G| 39.2 min|
> |DeepASA (w/o ATST)|**18.5**|**18.2**|**15.7**|**17.5**|**2.9 M**|**77.3 G**|**26.8 min**|
>
> Additionally, the benchmark for the STARSS23 dataset in Table 4 allows the use of external datasets and pre-trained models, following the criteria of the previous DCASE challenge. The models that achieved the best performance before DeepASA (NERC-SLIP, EINV2 with data augmentation chain) similarly performed data augmentation using large datasets like AudioSet and FSD50K or pre-trained their models for fine-tuning on the STARSS23 dataset. Consequently, the comparison in Table 4 can be considered fair, and the reason for increasing the number of foreground sources in ASA2 from 4 to 5 is specifically for fine-tuning on STARSS23.
>
> **Weak-Q3.** Foreground sources are fixed to J=5, however, some analysis regarding possible performance degradation when this assumption is violated would be helpful. Some analysis varying the degree of polyphonicity is expected. What is the computational and memory scaling behaviour?
>
> **A.** We conducted extra experiments with a larger number of sources than that assumed in the model (five). There is inherent performance degradation due to an increased number of sources, but the unseparated source signals are mostly merged into a single track without severely affecting the other four tracks. We will add this important discussion to the revised manuscript.
>
> (1) When the number of sources is increased to 6 and 7, there is a significant decrease in USS and SELD performance. This is obvious because (A) separating a larger number of sources is inherently difficult, (2) the model is designed to separate five sources, so unseparated objects are merged in some of the separated signals.
>
> Results for changing the number of sources (S)
> ||SI-SDRi↑|SDRi↑|ER↓|F1↑|LE↓|LR↑|SELD↓|
> |-|-|-|-|-|-|-|-|
> |S<=5|**11.0**|**11.7**|**28.8**|**70.2**|**18.5**|**76.9**|**0.230**|
> |S=6|2.4|5.9|27.7|48.5|40.3|54.5|0.418|
> |S=7|0.2|4.5|59.0|36.6|51.7|42.4|0.521|
>
> (2) However, this does not mean the total collapse of the model. When we evaluate the performance of individual sound objects, there is a big performance gap between the top-4 performance objects and the rest. This result stresses that four object signals are relatively well separated, whereas the rest objects are mapped to one remaining track.
>
> Detailed results for S=6 and 7
> |||SI-SDRi↑|SDRi↑|ER↓|F1↑|LE↓|LR↑|SELD↓|
> |-|-|-|-|-|-|-|-|-|
> |S=6|Top 4 average|**7.0**|**9.6**|**38.2**|**61.2**|**25.6**|**66.7**|**0.311**|
> ||Bottom 2 average|-6.8|-1.5|69.1|23.1|69.7|30.1|0.636|
> |S=7|Top 4 average|**6.6**|**9.1**|**38.9**|**60.6**|**29.2**|**64.5**|**0.325**|
> ||Bottom 2 average|-8.3|-1.6|85.8|4.6|81.7|12.9|0.784|
>
> (3) Computational and memory scaling behaviour
> - To train the model separating a larger number of sources, the output dimension of the object splitter must be increased. In contrast, the parallel decoders are parameter-shared, so the parameter size remains the same and the computational complexity only increases linearly.
>
> - The increased computational load and memory we measured are as follows. In the main paper, the training was performed with a batch size of 2. However, due to the memory limit of our GPU, the batch size was changed to 1 to measure memory size and runtime.
>
> Complexity comparisons for changing the number of sources
> ||Param.|MAC/s|GRU memory|runtime (/epoch)|
> |-|-|-|-|-|
> |S<=5|**105.0 M**|**99.1 G**|**12.3 GB**|**60.2 min**|
> |S=6|105.1 M|104.4 G|12.6 GB|62.3 min|
> |S=7|105.1 M|109.9 G|12.9 GB|64.1 min|
>
> **Minor weak-Q1.** The relationship between different loss components and their relative weighting could be better motivated in the main paper than in the appendix.
>
> **A.** In the main paper, lines 216-223, we have explained the weights of the loss functions and their relationships. We will also include additional information regarding permutation invariant training (PIT).
>
> **Question-Q1.** Could authors provide evidence for true object correspondence? .. some validation that object features actually correspond to individual sound sources rather than just being abstract encoded representations.
>
> **A.** The decoders processing different object features are identical parameter-shared decoders. If object features of individual sound objects include similar abstract representations of multiple objects, the outputs from the decoders should be the same.
> - In detail, the object splitter splits features for J objects and one noise. At this point, the size of the object feature map is (B, JxD, T, F), and to process each object in parallel, the object dimension J is reshaped into the batch dimension, resulting in (JxB, D, T, F). In other words, because a parameter-shared decoder is applied to the batch data, the decoder cannot be structurally tailored to a particular object. Our unclear illustration in Figure 4 might have caused this confusion, so we will amend the figure to clarify this mechanism.
>
> **Question-Q2.** Is it possible to show that each of the J=5 object features consistently tracks the same sound source across time? When sounding objects enter and exit a scene, are object features able to maintain semantic coherence?
>
> **A.** To verify this, we conducted a performance analysis on samples with an offset of more than 1 second during the occurrence of audio sources. The experimental results show that, although the performance of samples with an offset of more than 1 second decreases, the difference is not significant. This indicates that the proposed DeepASA can detect the same sound object through spatial and frequency information, even when there is an offset between the occurrences of the sources.
>
> Influence of the offset inserted between source onsets/offsets
> ||SI-SDRi↑|SDRi↑|ER↓|F1↑|LE↓|LR↑|SELD↓|
> |-|-|-|-|-|-|-|-|
> |Total|**11.0**|**11.7**|**28.8**|**70.2**|**18.5**|**76.9**|**0.230**|
> |with offset (> 1 s)|9.3|10.5|31.7|68.3|19.7|74.5|0.250|

---

### Official Review · Reviewer_gK5M · 2025-07-09

**Clarity:** 2
**Significance:** 3
**Originality:** 2
**Rating:** 4
**Confidence:** 5

**Summary:**

This paper proposes DeepASA, which trains a single deep neural network (DNN) to do source separation, dereverb, sound event detection, audio tagging, and direction-of-arrival (DOA) estimation.

This is realized by first implicitly separating the mixtures, creating hidden representations for each sound object, and then using various decoders on the hidden representations to estimate auditory parameters such as sound event and DOA.

Evaluation results on their own customized training and test dataset (i.e., ASA2) and two public datasets (MC-FUSS and STARSS23) show that the proposed method can produce strong performance for sound separation and auditory parameter estimation.

**Questions:**

1.	In Section 2.2, how is the synthesis window in inverse STFT computed?

2.	Line 146, what is “a single-channel object activation (onset and offset) curve”. From Figure 4(b), its dimension is JxT’ (without dimension C, so it is a label for all the sources combined). Is it a binary vector for each source, denoting when each source is active? It is really difficult to understand and this makes the reviewer unable to fully follow the “SED decoder” part, please explain and improve.

3.	Line 150, “During evaluation, only the predicted class probability and object activation curve are used to calculate the SELD scores.” Can the authors explain how? The predicted class prob has a dimension of JxC, and the object activation curve has a dimension of JxT’, according to Figure 4(b).

4.	Line 152, “The SED decoder combines a pre-trained audio teacher-student transformer (ATST) [12] with two separate convolutional recurrent neural network (CRNN) [11]-based branches summarizing time and frequency information.”, how is CRNN reflected in Figure 4(b)? The reviewer only notices AIST, but not CRNN. Are all the compared benchmark systems using the AIST model, which is trained on AudioSet, a much larger scale dataset than ASA2, MC-FUSS and STARSS23? If not, the comparisons in the experiments may not be fair at all.

5.	Line 114, “from 2M to C” should be “from 2M to D”? as C is defined as the number of sound types the model can handle.

6.	Figure 5, what is symbol “S”? Why the embedding after SED Decoder 1 is SC and why the embedding after DoA Decoder 1 is 3S? Are there any motivations of choosing the embedding dimension?

7.	Figure 5, in SED query branch, the attention output should be 3SxT’? Then why is it followed by a linear layer SC->D, where the dimension does not match.

8.	Figure 4(b), “SED decoder” contains multiple outputs including estimated activations, predicted classes and predicted SED. In Figure 5, “SED Decoder 1” outputs a SCxT’ tensor. How is this performed?

9.	Line 197, is ASA2 built upon the dataset proposed in [38]. From Section 3.1, this is not clear.

10.	Line 211, “The initial learning rate 4 ×10−4 was used for the Adam optimizer during pre-training, and 4 ×10−4 and 5 ×10−5 were used for fine-tuning to the MC-FUSS and STARSS23 datasets, respectively.” How is pre-training performed? So far, the authors talk nothing about pre-training. Do the authors first train DeepASA on ASA2, and then fine-tune the trained model on MC-FUSS and STARSS23? If this is the case, are all the compared benchmarks using similar ways and trained on the same training data?

11.	Line 221, what is “timestamp decoder”? this is a new thing that has not been described. It is not shown in Figure 4(b).

12.	Line 221, any motivations on using MSE loss for sound event detector?

13.	In Figure 4(b), what loss function is used for the audio decoder part? PIT? There is no description, and this is an important detail.

14.	Section 2.5, SED decoder has 3 outputs, which themselves may not be aligned with each other. Would this be possible and if so, how is this solved? and, in TCM, how the authors align DOA results with each of the 3 outputs in the SED decoder?

**Ethical Concerns:**

["NO or VERY MINOR ethics concerns only"]

**Final Justification:**

I'd like to thank the authors for addressing my comments and providing additional experiments. Many of my concens are addressed and I'd like to increase my rating as boardline accept.

**Limitations:**

yes

**Paper Formatting Concerns:**

See some concerns in my questions to the authors.

**Quality:**

2

**Strengths And Weaknesses:**

Strengths:

DeepASA can simultaneously deal with multiple tasks, and obtains reasonable improvement over existing methods.

Weaknesses:

1.	Title, “one-for-all” is not accurate, as auditory scene analysis contains many more tasks, while DeepASA only addresses a subset of them. In addition, the considered combination of tasks is not so novel, as compared with existing SELD tasks (which already deal with SED, audio tagging and DOA estimation simultaneously), DeepASA only introduces separation.

2.	The evaluation is only performed on simulated datasets, and lacks convincing evaluation results on real-recorded datasets for sound separation. On real-recorded data, sound separation is usually difficult, as real data often mismatches simulated data. In this case, would the method still perform well, considering that the method relies so much on obtaining good hidden representations for each sound object (i.e., if the hidden representations for each sound object is not good on, e.g., real-recorded data, then auditory parameter estimation would be also not good)?

3.	Section 2.2, better to use “analysis window” to replace the word “window”, as “window” is overloaded with many meanings in signal processing.

4.	“SED decoder” in Section 2.4 and Figure 4(b) feel messy and are difficult to follow. The writing needs improvement.

5.	Many details are missing regarding the model. See my detailed questions to the authors.

6.	The comparison with existing models in Table 3 and 4 are not so fair, as they are trained on different sets of data, and DeepASA uses the ATST model (which is pre-trained on a very large dataset). It might not be surprising that DeepASA can obtain better performance. See also my detailed questions.

---

> ### Author Rebuttal · Authors · 2025-07-31
>
> **Weakness**
>
> **A1.** Thank you for the reviewer’s comment. We will change the title of the manuscript to “DeepASA: An object-oriented multi-purpose network for auditory scene analysis.”  Additionally, the ‘separation’ task of our work represents ‘multichannel-to-multichannel separation’ and ‘dereverberation’, which involves separating the direct and reverberant waveforms in a multichannel format, unlike conventional separation models that produce single-channel output.
>
> **A2.** The STARSS23 dev-test dataset is a real-world dataset captured in a real environment. So the SELD performance has been validated on a real dataset. For the USS task, we conducted an additional experiment by fine-tuning the pretrained DeepASA over datasets generated from real-world room impulse responses (RIRs). The fine-tuned model was then evaluated by unseen real-world RIRs.
>
> Comparison between datasets with simulated and real-world RIRs
> ||SI-SDRi↑|SDRi↑|ER↓ |F1↑ |LE↓|LR↑|SELD↓|
> |-|-|-|-|-|-|-|-|
> |simulated RIR|11.0|11.7|28.8|70.2|18.5|76.9|0.230|
> |real RIR|10.6|11.4|32.7|66.4|13.4|72.3|0.254|
>
> The experimental results show a slight decrease in USS performance, as well as in most of SELD performance. As the reviewer mentioned, this performance degradation can be attributed to the discrepancy in the hidden representation of simulation and real RIRs. Ideally, training with a large set of RIRs, including both simulated and measured RIRs, would improve the generalizability of models.
>
> **A3.** We will use “analysis window” instead of “window.”
>
> **A4.** According to the reviewers’ comments, we will amend the explanation of “SED decoder” in Section 2.4 as follows.
> “The SED decoder decodes the object features of each sound source into the predicted class probability (1xC), the binary object activation curve for event onset and offset (T’x1), and the predicted SED map (T’xC). Here, T’ and C denote the number of time frames and the number of classes, respectively. The predicted SED map is only used in the training stage to guide the joint detection of a sound event and class. In the inference stage, we use the SED map produced by multiplying the predicted class and activation curve, resulting in a single SED map for each sound source. This is to exclude the possibility of more than one class being mapped to the SED map for each sound object.”
>
> **A5.** We will update the text and figures (see below).
>
> **A6.** The reviewer’s comments can be summarized as (1) Use the same training dataset for comparison, and (2) The large size of the pretraining dataset for the ATST (SELD task).
> As for the comment (1), the objective of the comparisons is to demonstrate the “generalizability” of the proposed model pretrained on the multi-task dataset. The “fair comparison using the same dataset” is separately studied in the table below, where the same dataset (ASA2 dataset) was used and compared to the SOTA USS model (DeFT-Mamba-MISO). For the case when using only MC-FUSS, we conducted additional experiments, demonstrating that the proposed model still outperforms DeFT-Mamba even when trained from scratch without pretraining.
>
> Results on the MC-FUSS dataset (SI-SDRi)
> ||| S=2  | S=3  | S=4  | Total |
> |-|-|-|-|-|-|
> |pre-trained on ASA2|DeFT-Mamba| 18.4 | 17.2 | 14.1 | 16.6  |
> ||DeepASA (w/o ATST)| 18.8 | 19.0 | 17.4 | 18.4  |
> ||DeepASA (w/ ATST)|18.9|19.1|17.6|18.5|
> |from scratch|DeFT-Mamba|18.4|17.1|13.8|16.4|
> ||DeepASA (w/o ATST)|18.5|18.2|15.7|17.5|
>
> For comment (2), the SELD models in Table 4 were also trained using the large auxiliary datasets. The DCASE 2023 challenge allowed the use of external datasets and pre-trained models, so ‘NERC-SLIP’ and ‘EINV2 with data augmentation chain’ were also trained by large auxiliary datasets like AudioSet, FSD50K.
>
> **Questions**
>
> **A1.** We used a fixed Gaussian window with a zero mean and standard deviation of 6.67 ms for the synthesis window in inverse STFT.
>
> **A2.** Thank you for pointing out this ambiguity in the figure. SED decoders for “J” different objects share the same architecture and weights (parameter-shared), and this is expressed as the batch dim. Accordingly, for each object, the object activation curve is a single-channel binary vector across time (1xT'). To avoid confusion, we will revise Fig. 4(b) with a clear presentation of this parameter-shared architecture.
>
> **A3.** For each object in the batch(J), a T'xC-sized SED map is produced from the estimated activation curve and the class probability vector (please refer to W-A4). The SED maps for J objects are concatenated to form the output of size JxT'xC. We will reflect this detailed explanation in the revised manuscript along with the modified Fig. 4(b).
>
> **A4.** “CRNN” denotes multiple repetitions of Conv2d and Maxpool2d, followed by GRU and FC layers in Fig. 4(b). Specifically, among the three red branches of the SED decoder, the middle path with seven Conv+Maxpool layers outputting (JxT'xD) is the T-CRNN path, and the bottom path with two Conv+Maxpool layers outputting (JxF'xD) corresponds to the F-CRNN path. We will revise Fig. 4(b) to enhance its clarity.
>  Again, Tables 3 and 4 are for demonstrating the generalization capability, and the architectural performance comparison under the same training dataset should be referred to Table 1.
> - The other models compared in Table 4 do not use the pre-trained ATST, but top-rank models (NERC-SLIP, EINV2 with data augmentation chain) also use large datasets (AudioSet, FSD50K) or pre-trained models. This is because the STARSS23 dataset is somewhat small to achieve robust classification across diverse classes of data.
> - The other models in Table 3 also do not utilize the ATST, but the use of ATST is less relevant for the USS task because the SED decoder is removed in fine-tuning for the USS task.
> -  For showing the architecture-originated difference only, we will provide the results of DeepASA trained solely on each dataset without using the pre-trained ATST or any pre-training on ASA2, in which DeepASA still outperforms existing SOTA models.
>
> ASA2 dataset result comparison between DeepASA with and without ATST
> |DeepASA|SI-SDR↑|SDRi↑|ER↓|F1↑|LE↓|LR↑|SELD↓|Param.|MAC/s|
> |-|-|-|-|-|-|-|-|-|-|
> |w/ pre-trained ATST|11.0|11.7|28.8|70.2|18.5|76.9|0.230|105.5M|99.1G|
> |w/o ATST|11.0|11.7|30.1|69.5|18.5|74.8|0.240|8.2M|91.0G|
>
> MC-FUSS dataset result comparison with DeepASA without ATST (trained from scratch)
> ||S=2|S=3|S=4|Total|Param.|MAC/s|Runtime|
> |-|-|-|-|-|-|-|-|
> |DeFT-Mamba|18.4|17.1|13.8|16.4|3.6 M|83.8 G|39.2 min|
> |DeepASA (w/o ATST)|18.5|18.2|15.7|17.5|2.9 M|77.3 G| 26.8 min|
> |DeepASA (w/ ATST)|18.9|19.1|17.6|18.5|2.9 M|77.3 G|26.8 min|
>
> **A5.** Thank you for pointing out the typo. We will change “from 2M to C” into “from 2M to D.”
>
> **A6** S is equivalent to J (maximum number of sound sources). We will change S to J in Fig. 5 and will use S to denote the ‘actual’ number of sources in the mixture.
> - For each sound object, the SED decoder and DoA decoder output T'xC and T'x3 (3: X, Y, Z coordinates) size tensors, respectively. These are concatenated for J sound objects and then transposed to JCxT' and 3JxT' tensors.
> - In TCM, the embedding dim. is T' and the sequence dim. is JC or 3J. This is for emphasizing channels showing similar temporal activities across SED and DoAE through the cross-attention. After the cross-attention, the output is transposed again to make T' the sequence dim.
>
> **A7.** In the SED query branch, the query is the SED output (JCxT'), and the key and value are the DoA output (3JxT'). The output size of the attention follows that of the query, followed by transpose; it becomes T'xJC, and then the channel dim. is changed from JC to D through a linear layer (size: T'xJC → T’xD).
>
> **A8.** The SED decoder outputs three outputs, but only the SED map is used for training Net 2. As answered in replies W-A4 and Q-A3, J SED maps with a size of T'xC are produced and processed to form a T'xJC tensor in Fig. 5.
>
> **A9.** ASA2 is based on the ASA dataset, but we modified it extensively to enable the comprehensive tasks discussed in this study. ASA2 supports (1) the separation of direct and reverberation components (for dereverberation), (2) MIMO separation, as compared to MISO separation of the ASA dataset, and (3) an increased maximum number of foreground sound sources, from four to five. We will add these details to the appendix.
>
> **A10.** DeepASA's pre-training was conducted on the ASA2 dataset, followed by fine-tuning on the datasets for each downstream task. We will add a detailed explanation in the manuscript.
> Models in the MC-FUSS dataset benchmark did not involve pre-training, whereas some models for the STARSS23 included pre-training and data augmentation on large datasets (AudioSet, FSD50K). To show the influence of pretraining a USS model on a large dataset, we will add the experimental results presented in replies W-A6 & Q-A4 to the manuscript.
>
> **A11.** It is another consistency fault. We will unify the terminology to "object activation curve estimation" and also revise other parts with that expression (Line 163 and Line 221).
>
> **A12.** We used MSE because it is a typical choice in previous works for SED. (“CSTFormer” (Y. Shul, et. al), “Ensemble of ACCDOA and EINV2” (K. Shimida, et. al,))
>
> **A13.** The SA-SDR was used as a loss for the audio decoder, which is mentioned in line 216. Also, the PIT technique was used, and we will mention this in the revised manuscript.
>
> **A14.** As mentioned in the previous reply, the SED decoder outputs three components, but ultimately, the SED map of T'xC is constructed for each object from the predicted class probability (1xC) and object activation (T'x1). DoA decoder also generates T'x3 output for each object. The alignment between the SEDs and DoAs is accomplished in TCM through the cross-attention by assigning the time dim. as a sequence dim. We will add this explanation regarding this part in the manuscript and revise Fig. 4(b) and 5 accordingly.

---

> > ### Comment · Reviewer_gK5M · 2025-08-09
> >
> > I'd like to thank the authors for addressing my comments and providing additional experiments. Many of my concens are addressed and I'd like to increase my rating as boardline accept.

---

### Note · Authors · 2025-08-11

Dear reviewers and AC,

The reviewers' valuable comments have been helpful in improving the quality of the research. If the paper is accepted, we will make sure to incorporate the discussed points into the camera-ready version.

---

### Decision · Program_Chairs · 2025-09-17

**Decision:**

Accept (poster)

**Comment:**

This paper proposes DeepASA, which trains a single deep network to perform multiple audio scene analysis tasks, including source separation, dereverb, sound event detection, audio tagging, and direction-of-arrival estimation. This is realized by first separating the mixtures (Object-Oriented Processing strategy), creating hidden representations for each sound object, and then using various decoders on the hidden representations to estimate auditory parameters, with further refinement achieved by a Chain-of-Inference mechanism. Through extensive discussions the reviewers are happy with the method and results presented in this paper.